# Empathi: embedding-based phage protein annotation tool by hierarchical assignment

Alexandre Boulay [1,2] ✉, Audrey Leprince [1], François Enault [3], Elsa Rousseau [2,4,5,6] & Clovis Galiez [7] ✉

Bacteriophages, viruses infecting bacteria, are estimated to outnumber their cellular hosts by 10-fold, acting as key players in all microbial ecosystems. Under evolutionary pressure by their host, they evolve rapidly and encode a large diversity of protein sequences. Consequently, the majority of functions carried by phage proteins remain elusive. Current tools to comprehensively identify phage protein functions from their sequence either lack sensitivity (those relying on homology for instance) or specificity (assigning a single coarse grain function to a protein). Here, we introduce Empathi, a protein-embedding-based classifier that assigns functions in a hierarchical manner. New categories were specifically elaborated for phage protein functions and organized such that molecular-level functions are respected in each category, making them well suited for training machine learning classifiers based on protein embeddings. Empathi outperforms homology-based methods on a dataset of cultured phage genomes, tripling the number of annotated homologous groups. On the EnVhogDB database, the most recent and extensive database of metagenomically-sourced phage proteins, Empathi doubled the annotated fraction of protein families from 16% to 33%. Having a more global view of the repertoire of functions a phage possesses will assuredly help to understand them and their interactions with bacteria better.

Bacteriophages or phages—viruses that infect bacteria—are some of the most abundant biological entities on earth, being present everywhere from the ocean and the soil to our very own bodies[1–3]. Despite this, until recent years, phages have been overlooked in studies of the microbiome that have mainly focused on the bacterial component. Furthermore, wet-lab studies on phages are slow and labor intensive, traditionally requiring phages to be cultured in the presence of a known bacterial host. The development of next-generation whole-metagenome shotgun sequencing methods, has substantially accelerated the study of phages, allowing to sequence them directly in their natural habitats, thereby circumventing phage culturing. However,

being able to assemble new phage genomes from massive metagenomics sequencing data comes with the challenge of characterizing these highly diverse phages and the proteins they are composed of, as well as determining their host.

Phage protein function prediction is a major challenge. Using EnVhogDB[4], the most recent and extensive database of phage proteins collected from metagenomic sources, it was shown by Pérez-Bucio et al. that only 16% of the diversity of phage protein families has been assigned a function. This issue has been the focus of numerous studies[5–23], most of them limiting their efforts to predicting one type of protein at a time. The most studied of these are phage virion proteins

[1]Département de biochimie, de microbiologie et de bio-informatique, Université Laval, Québec, QC, Canada. [2]Centre Nutrition, Santé et Société (NUTRISS), Institute of Nutrition and Functional Foods (INAF), Université Laval, Québec, QC, Canada. [3]Université Clermont Auvergne, CNRS, LMGE, Clermont-Ferrand, France. [4]Département d'informatique et de génie logiciel, Université Laval, Québec, QC, Canada. [5]Centre de Recherche en Données Massives de l'Université Laval, Québec, QC, Canada. [6]Institut Intelligence et Données (IID), Université Laval, Québec, QC, Canada. [7]Université Grenoble Alpes, CNRS, Grenoble INP, LJK, Grenoble, France. ✉e-mail: alexandreboulay@outlook.com; clovis.galiez@univ-grenoble-alpes.fr

(PVPs), the structural proteins of phages[5–17,24–27]. In particular, receptor binding proteins (RBPs) that allow binding and recognition of the bacterial host[18,19], and lysins that degrade the peptidoglycan or exo-polysaccharide layers surrounding and protecting bacteria[20–23] have received significant research attention.

Currently, the most widely used method for assigning general functional annotations to phage proteins is an approach based on profile hidden Markov models (pHMMs) which relies on sequence homology[28]. Yet, as pointed out by Flamholz et al.[29], in viral metagenomics, these methods are constrained by the limited number of annotated proteins that can be used to construct probabilistic sequence models and by the large diversity of protein sequences.

Recently, alignment-free methods based on protein embeddings, fixed-size real-valued vectors (Fig. 1b) obtained from protein language models (PLMs) such as ProtTrans[30] and ESM2[31], were developed. These models have been trained on an enormous corpus of protein sequences and have been demonstrated to capture the structural and functional information from protein sequences[29,32].

Different tools based on protein embeddings were recently developed for phage protein function annotation[6,19,20,22,29], with all but one being specific to particular protein families. For example, Yang et al.[19] trained a classifier to identify tail-spike proteins with a distinctive beta-helix domain, and Concha-Eloko et al. proposed DepoScope[22], a tool trained for depolymerase detection and functional domain identification. The only classifier able to predict general phage protein functions, VPF-PLM[29], is a multilabel classifier that uses embeddings obtained using ESM2 and trained to predict the basic PHROG[33] categories ("tail", "head and packaging", "connector", "lysis", "transcription regulation", "integration and excision", "DNA, RNA and nucleotide metabolism", "moron, auxiliary metabolic gene and host takeover" and "other"). However, these categories possess overlapping molecular-level functions (for example, between the "DNA, RNA and nucleotide metabolism" and "transcription regulation" categories), creating noise in the training data that hinders the accuracy (global model performance) and sensitivity (i.e., how well the model predicts positive instances) that can be achieved by models trained on them.

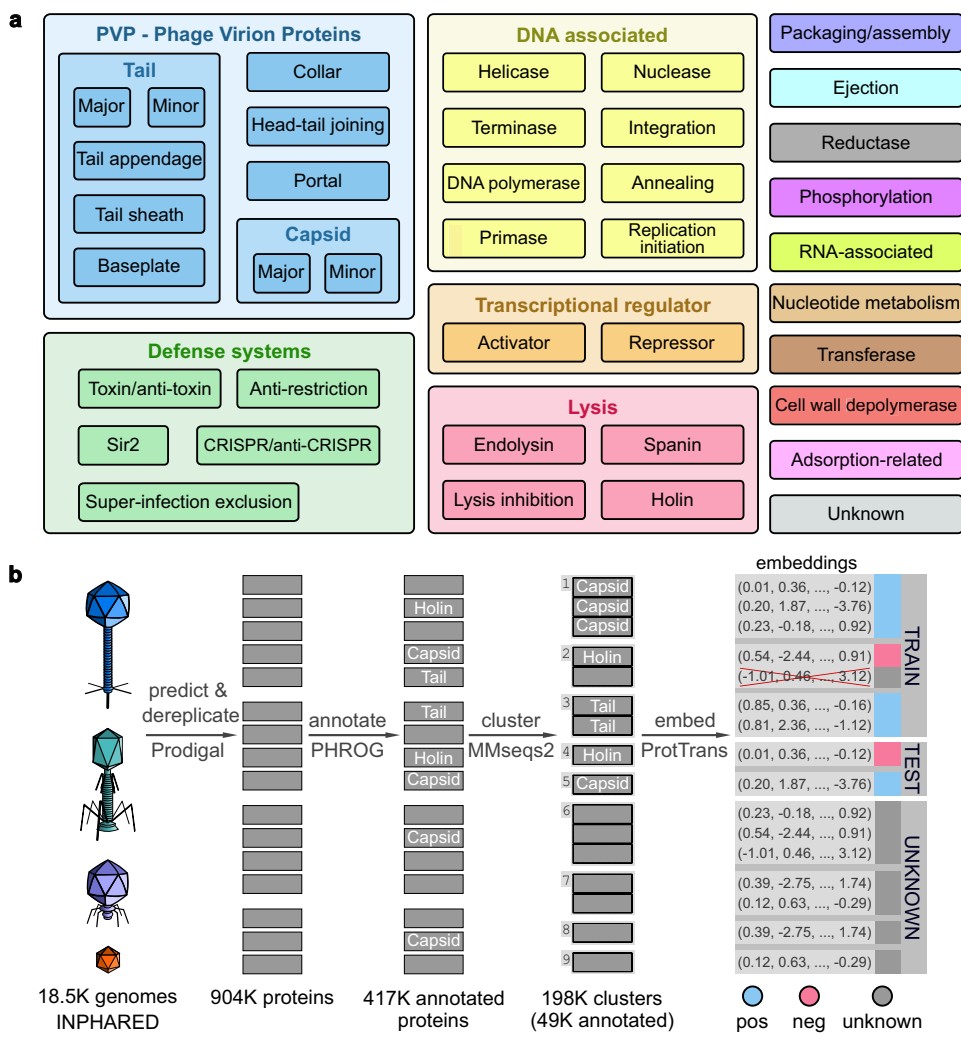

**Fig. 1 | Definition of Empathi categories and dataset preparation. a** Definition of the hierarchical functional categories used in Empathi. PHROG annotation terms are placed in all categories they fit in (e.g. "tail protein with lytic activity" is placed in tail, pvp and cell wall depolymerase). The unknown label is given to proteins not assigned to any category by the Empathi models. The colors defined here for each category were used in all following figures. **b** Dataset preparation for machine learning. Phage genomes were downloaded from public databases using INPHARED. Protein sequences were predicted using Prodigal and annotated by sequence-pHMM comparison with proteins of known function in the PHROG database. Proteins were clustered using MMseqs2 and embedded using ProtTrans to obtain embeddings 1024 values long. To train models, all proteins with known functions from one cluster were either placed in the training set or in the testing set. The positive and negative labels were defined independently for each binary model that was trained; in this example, the positive set is composed of phage virion (structural) proteins.

In this paper, we present Empathi, an Embedding-based Phage Protein Annotation Tool that Hierarchically assigns protein functions. To this end, a hierarchical scheme for functional categories that respects molecular-level functions was defined from the PHROG classification[33] to be better adapted for machine learning (ML) classification. Empathi is composed of multiple binary models trained on proteins from completely sequenced phages. These models are then used to assign functions to new proteins, starting from general annotations such as structural or DNA-associated to more precise functions when possible. Empathi significantly outperforms homology-based methods, tripling the number of annotated homologous groups in our original dataset of cultured phage genomes. We demonstrate that our approach can be used on new phage genomes (from recently published viromes), showing consistency of predictions through genomic maps that illustrate the colocalization of protein functions and that demonstrate our tool's pertinence in comparison to state-of-the-art approaches such as PHROG pHMMs (homology identification), VPF-PLM (protein embeddings) and PHOLD[34] (protein structures). Finally, Empathi was employed to increase the proportion of annotated clusters in EnVhogDB from 16% to 33% and in EFAM[35] from 34% to 58%.

## Results

### New hierarchical scheme for functional groups

PHROG categories are not adapted for machine learning classification. Indeed, many PHROG categories encompass various molecular-level functions. For example, the head and packaging category is composed of structural proteins, internal proteins with lytic domains and terminase proteins that can bind DNA. Furthermore, the same molecular-level function is found in various PHROG categories. For example, DNA-associated proteins can be found in 7 out of the 9 PHROG categories.

Here, from a biological perspective and with machine learning purposes in mind, similar PHROG annotation terms were grouped together into new functional categories that respect molecular functions (see Fig. 1a). The complete list of PHROG annotation terms constituting each newly defined functional category can be found in Empathi's code repository in data/functional_groups.json. These newly defined functional categories include groups such as baseplate proteins, nucleases and adsorption-related proteins, that are, when possible, classified into more general ones (PVP, DNA-associated, lysis-associated). Proteins can be associated with multiple categories. For example, tail proteins with lytic activity are included in the PVP category and in the cell wall depolymerase category.

### Building and testing models

As visualized in Fig. 1b, 904k dereplicated proteins were obtained from 18.5k phage genomes downloaded from GenBank using INPHARED[36]. Almost half, 417k proteins (46%), were placed in at least one of the 44 newly defined functional categories based on their sequence similarity to PHROG pHMMs. This annotation pipeline makes it possible to readily update our models with data from new genomes in GenBank or from other sources, but is limited, for now, to proteins from cultured genomes to ensure the good quality of annotations. All 904k proteins were clustered at 30% sequence identity and 80% coverage into 198k clusters using MMseqs2[37]. A quarter of these, i.e., 49k clusters, contained annotated proteins. These clusters were used to create the training and testing sets in order to reduce data leakage due to homology.

A binary model was trained for each of the 44 functional categories using 80% of clusters and tested on the remaining 20% of clusters (Supplementary Table 1). For example, one model was trained to identify PVPs from non-PVPs (i.e., from all other proteins). Training a binary model for each category separately, using specifically designed training sets, ensures that as little noise is present as possible to influence models (see the "Methods" section for details). The precision (proportion of correct positive predictions; higher precision indicates lower false positive rate), sensitivity (how well the model predicts all positive instances; higher sensitivity indicates lower false negative rate) and F1-score (harmonic mean between precision and sensitivity) were analyzed to assess the performance of the models. The F1-score for all binary models except one was greater than or equal to 88% with three quarters of models reaching scores of at least 95% (Supplementary Table 1). Only the model trained to predict collar proteins was less performant, achieving a score of 60% (Supplementary Fig. 1 and Supplementary Table 1). This is most likely due to the very limited number of collar proteins in the dataset (233 proteins corresponding to 39 clusters in the training set). In consequence, the collar protein model was removed from the final version of Empathi. The precision and sensitivity curves demonstrate that models are confident in their predictions, being able to achieve an excellent sensitivity (83–100% at 50% confidence) whilst conserving an excellent precision (80–100% at 50% confidence) (Fig. 2a, Supplementary Fig. 1 and Supplementary Table 1).

Many of the errors made by the models stem from biologically similar functions. The baseplate model classified tail appendage proteins as baseplate proteins and vice versa reflecting the biological similarity of these proteins. Similarly, the primase model incorrectly classified some helicases. In addition, some proteins annotated only as tail proteins are predicted by our model as being adsorption-related proteins. These are likely not errors, but the protein annotations are too general to validate our predictions.

### Evaluation of robustness

Even though proteins detected as similar by sequence-sequence comparisons were separated into either the training or the testing set, remote homologs whose similarity is not detected by MMseqs2 might be present in both sets. As pHMM-pHMM comparisons of the PHROGs detect more distant homologous relationships between proteins, new training sets were built by removing one PHROG as well as any PHROG similar to it. In this manner, retrained models were tested on proteins that had no resemblance, even remote, to any protein in the training dataset.

Overall, models predicted the function of interest accurately for these holdout PHROGs (Fig. 2b), having an F1 score greater than or equal to 75% in 51 of the 60 replicates (85%). Importantly, a drop in sensitivity but not in precision is usually observed. Lower model performances on some holdout groups correspond to cases where the holdout group was very large and where all examples of a specific type of protein (e.g., MotB-like transcriptional regulator) have been removed from training by the holdout procedure.

This test serves to push the boundaries of our classifiers. It demonstrates that for the most part, even if we remove all similar proteins—even those remotely related (as far as pHMM-pHMM comparisons can detect)—from the training data, our models remain consistent and able to generalize. It highlights their ability to identify a functional signal within the protein embeddings more sensitively than by using HMM profiles.

### Expanding the proportion of annotated proteins

As many as 483k proteins (54%) in the INPHARED dataset, corresponding to 150k clusters (76%), were not similar to any PHROG pHMM with a known function (considering all PHROG hits with an e-value < 0.001). Additionally, 2.4k proteins had a hit to an unknown PHROG that itself had strong pHMM-pHMM similarity to another annotated PHROG (e-value < 0.001 and coverage >80% between similar PHROGs). Moreover, 13.9k unannotated proteins (2.4k clusters) were found in clusters containing other annotated proteins, resulting in 467.5k proteins (52%) and 148k clusters (75%) that are fully unannotated. Among these proteins that even sensitive similarity search methods have failed to functionally annotate, 61% (285k proteins) were assigned a function

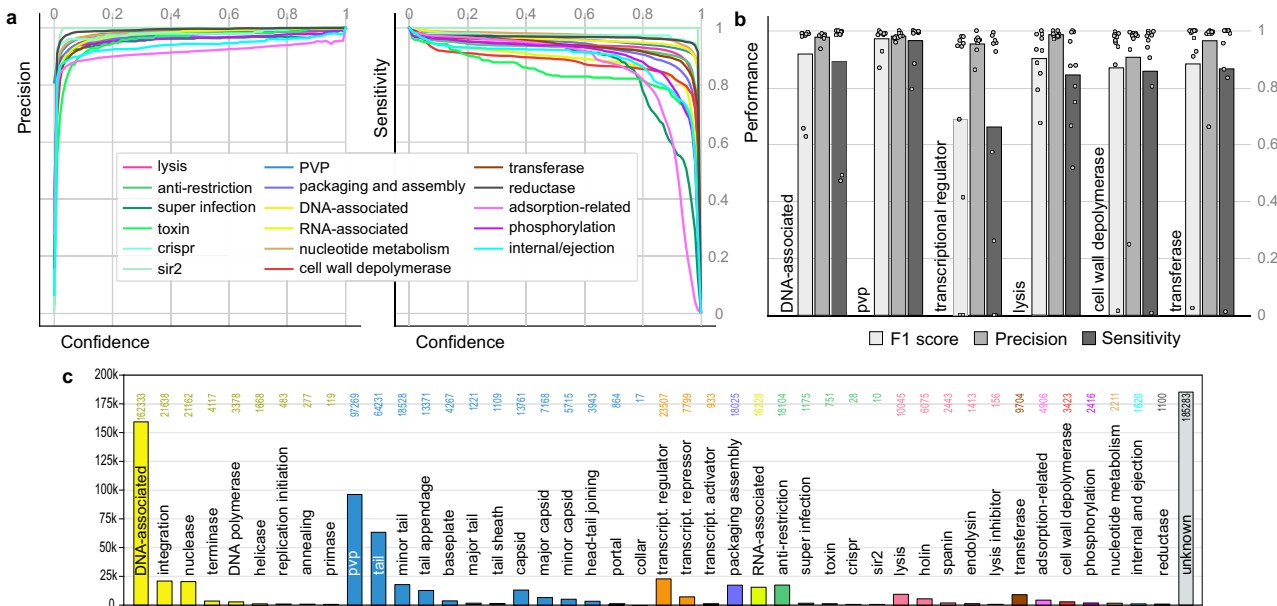

**Fig. 2 | Empathi performance and predictions for previously unannotated proteins. a** Precision and sensitivity curves as a function of confidence (prediction probability) for Empathi binary models trained on the general functional groups. PVP phage virion protein. **b** Robustness analysis: F1 scores, precision and sensitivity of models on holdout PHROGs using a confidence threshold of 50%. Bars represent the average performance over 10 iterations. As no transcriptional regulators were predicted in two iterations, the precision for these iterations could not be computed (zero division) and was thus not included in the average score. **c** Predicted protein counts for each functional group in the fraction of the original dataset (INPHARED) that did not align to proteins with known functions in the PHROG database.

with Empathi. In total, almost four phage proteins out of five now have a predicted function (718k proteins, 79% of the dataset), bringing the total ratio of annotated clusters from 25% to 73%. On average, the confidence of the highest-level prediction for previously unannotated proteins (in completely unannotated clusters) is 80.6%.

A high diversity of protein functions is observed in this previously unannotated fraction of the dataset (Fig. 2c). "DNA-associated" (35%) and "phage virion proteins" (21%) are the functions that were assigned to the most proteins. Among the predicted PVPs, about two-thirds are tail proteins. In addition, 23.5k transcriptional regulators, 21.6k integration-related, 21.2k nucleases, 18k packaging/assembly related, 16.2k RNA-associated and 18.1k anti-restriction proteins were also identified, highlighting the unexplored diversity of these proteins.

**Protein annotation of complete genomes**
Complete phage genomes were obtained from three recent viromes[38–40] sampled from various environments (human gut, sulfuric soil and marine water) and, for the purpose of this analysis, two complete genomes were randomly picked from each study (see https://doi.org/10.5281/zenodo.14036011 for their genomic sequences). As they each possess multiple predicted tail-associated proteins (Fig. 3), these phages likely correspond to the *Caudoviricetes* class.

Functions were assigned to the proteins from these genomes using Empathi, VPF-PLM[29] (based on protein embeddings), PHROG[33] pHMMs (based on sequence homology) and PHOLD[34] (based on predicted structures). A greater number of proteins were assigned a function using Empathi than the three other approaches. Out of a total of 574 proteins for the six genomes presented in Fig. 3, 414 proteins were assigned a function using Empathi, compared to 263 using PHOLD, 232 using PHROG pHMMs and 202 using VPF-PLM. For the most part, the annotations obtained from all four methods are coherent. For example, most transcriptional regulators, integration proteins, RNA-associated and nucleotide metabolism proteins from Empathi categories are consistent with the 'DNA, RNA and nucleotide metabolism' PHROG category, thus colored in yellow-orange in both

cases. The same is true for phage virion proteins (PVPs), packaging proteins and internal proteins from Empathi categories being consistent with the "tail", "head and packaging" and "connector" PHROG categories depicted in varying shades of blue. About 150 proteins received annotations from Empathi, but not from the other tools. Evaluating the false positive rate of these proteins is harder to do, as no annotation is available. However, a lower confidence is observed for 126/150 of these proteins, demonstrating that Empathi internally recognizes that these sequences are more distinct than those present in its training set.

A more in-depth analysis of the predictions made for one genome (PP079085.1) from Ni et al.'s study (Fig. 3) helps elucidate some of the observed differences (Supplementary Table 2). First, two proteins predicted as being lysis-associated proteins (see red arrows) by Empathi are classified under the head and packaging category by PHROG pHMMs (phrog_2860) and by PHOLD. The PHROG annotation terms, instead of categories, reveal these proteins are indeed endolysins. Furthermore, VPF-PLM also predicted them as being "head and packaging" proteins which is expected from a model trained on the PHROG categories, yet the most important annotation—the one related to the actual function of the protein—is lost. Second, there are three proteins predicted by Empathi as being internal/ejection proteins (in cyan) and these are indeed internal proteins placed under the 'head and packaging' category using PHROG pHMMs (phrog_308, phrog_418, phrog_6651). Third, two proteins predicted as DNA-associated and packaging-related (in yellow) by Empathi are a terminase small and large subunit which is also consistent (also in the 'head and packaging' PHROG category; phrog_2, phrog_11494). Finally, two glycosyltransferases (see brown arrows) are predicted as being transferases by Empathi, but placed in the 'moron, auxiliary metabolic gene and host takeover' PHROG category (phrog_14945, phrog_34859). Once again for these last three examples, VPF-PLM usually predicted the large category like with PHROG pHMMs and PHOLD, but loses critical information about the actual molecular function of those proteins.

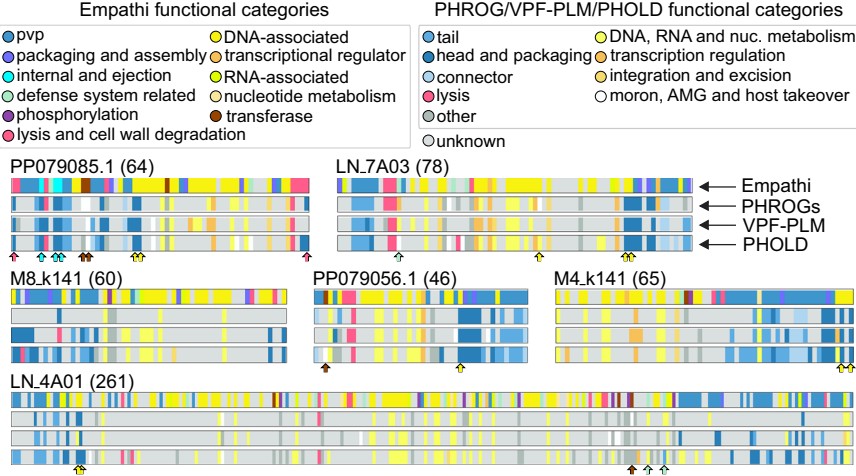

**Fig. 3 | Genomic maps of predicted functional annotations.** Functional maps of six genomes assembled from three viromes[38–40] colored according to the annotations obtained using Empathi, PHROG-pHMMs, VPF-PLM and PHOLD. Note that similar colors were chosen for corresponding functional categories between Empathi and PHROGs, and that each protein is represented with a uniform width in the genomic maps. Arrows indicate proteins for which the Empathi prediction corresponds to the more precise annotation term rather than category obtained using PHROG pHMMs or PHOLD. The number of proteins found in each genome is indicated in parentheses. pvp phage virion protein. PP079085.1 and PP079056.1 come from Ni et al., LN_4A01 and LN_7A03 come from Garmaeva et al. and M4_k141 and M8_k141 come from Bi et al.

Of note, similar functions assigned by Empathi seem to be highly colocalized. Phage virion proteins, DNA-associated proteins and lysis-associated proteins are usually grouped together. This is consistent with the fact that proteins having similar functions are usually colocalized in phage genomes in order to be expressed at the same time.

### Annotation of the EnVhogDB database

The EnVhogDB database being composed of proteins obtained from metagenomic datasets including EFAM, RefSeqVirus, IMG/VR and GL-UVAB, it is representative of a much greater diversity than the proteins used to train Empathi. Since proteins that have a more distant homology to proteins in the training set are more likely to lead to erroneous predictions, a more stringent confidence threshold of 95% was used to assign predictions using Empathi, resulting in an increase from 16% to 33% (17% increase) of annotated protein clusters in the whole dataset compared to previous annotations based on sequence homology. These can be visualized in Fig. 4. Once again, a great diversity of protein functions can be observed in the dataset, with phage virion proteins (mostly tail proteins) and DNA-associated proteins being the most abundant. Using Empathi's default confidence threshold of 50%, the proportion of annotated orthologous groups increased to 67.5%, therefore resulting in many more predictions but likely at the expense of a higher rate of false positives.

Finally, a random sample of 1000 proteins was taken from the previously unannotated fraction of the EnVhogDB database with the objective of comparing Empathi's and VPF-PLM's ability to assign predictions to new proteins. Empathi annotates 25.8% of proteins at a confidence of 95%, while VPF-PLM can only annotate 17.3% using its calibrated confidence thresholds (see calibrated thresholds in Flamholz et al.[29]). It is important to note that the confidence thresholds used by VPF-PLM to assign predictions are always much lower than the 95% used by Empathi for this analysis. In fact, large categories such as "DNA, RNA and nucleotide metabolism" and "head and packaging" use thresholds of 25% and 41% respectively.

### Annotation of the EFAM database

The EFAM[35] database is composed of 240,311 viral protein families collected from marine ecosystems. It was originally annotated with DRAM[41], a large-scale, multi-database, homology-based method that was shown by Zayed et al.[35] to "[double] the number of annotations obtainable by standard, single database annotation approaches" on the EFAM database, resulting in 80,431 annotated clusters (33.5%). Using Empathi, still with a confidence threshold of 95%, 138,785 clusters (57.7%) received an annotation, corresponding to a 24% increase compared to DRAM.

Complementary to DRAM annotations, 29,355 proteins (2529 clusters) in EFAM, for which standard homology-based methods failed to assign annotations, were previously annotated using virion-associated metaproteomic data. Empathi was used on these proteins specifically to evaluate its sensitivity on PVPs from an external dataset. Always using a confidence threshold of 95%, a total of 29,036 of these proteins (99%), corresponding to 2400 clusters, were predicted correctly as being virion-associated, while only 319 proteins (1%), corresponding to 129 clusters, were wrongly predicted.

Next, Empathi's consistency was tested by looking at the predictions made within each cluster of homologous proteins. Because these proteins are very similar, the predictions they receive from our model should also be similar. When looking at the 66,056 clusters previously annotated by DRAM that also received predictions from Empathi, 51,856 clusters (79%) indeed received identical predictions for every protein. A total of 1600 clusters (2%) received identical predictions with some proteins receiving no function. The proteins in 12,201 clusters (18%) were predicted as sharing at least one general function but were assigned either differing specific functions or a differing second general function. Finally, only 399 clusters (<1%) received differing predictions. When looking at the 72,729 clusters only annotated by Empathi, 44,669 clusters (61%) indeed received identical predictions for every protein. A total of 9995 clusters (14%) received identical predictions with some proteins receiving no function. The proteins in 17,138 clusters (24%) were predicted as sharing at least one general function but were assigned either differing specific functions or a differing second general function. Finally, only 927 clusters (1%) received differing predictions. There is thus a tendency in the previously unannotated portion of the dataset for clusters to possess more unknown predictions from Empathi as well. Most importantly, very few clusters possess proteins with differing annotations.

### Discussion

In this work, we developed Empathi, a tool that leverages the highly informative representations generated by protein language models to

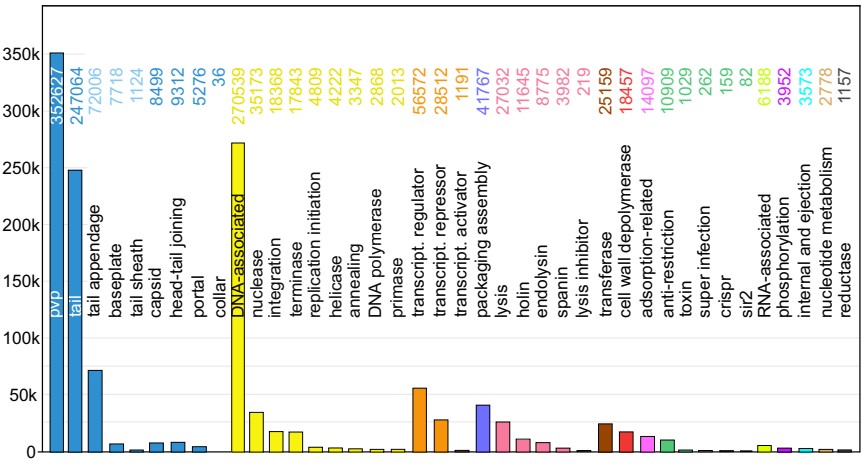

**Fig. 4 | Empathi predictions for the EnVhogDB database.** Predicted protein counts for each functional group at a confidence >0.95.

annotate beyond standard and remote homology. It constitutes a significant improvement from the most recent model proposed by Flamholz et al. for this task. By using binary models and reorganizing annotations to make them more consistent with the molecular functions of proteins, we are able to improve the accuracy and sensitivity of models trained to predict protein functions. Beyond experimental validation using the testing dataset, the high colocalization of predicted functions in complete genomes further demonstrates Empathi's consistency. Finally, Empathi was applied to the EnVhogDB database, doubling the proportion of annotated protein clusters from 16% to 33%, and to EFAM, increasing the proportion of annotated clusters from 34% to 58%.

Being constituted of independent binary models, Empathi can assign multiple functions to proteins, such as a specific and a general function like "nuclease" and "DNA-associated", or even multiple general functions. For example, some packaging proteins can also bind to DNA[42], and some structural proteins can also have lytic domains[20]. Tools only assigning a single annotation strongly limit subsequent analyses. Out of the 15.4k predicted cell wall depolymerases—including lysins and exopolysaccharide (EPS) depolymerases—in the EnVhogDB database, 3k are also predicted as being structural proteins. This is important as it hints to the biological role of these proteins. EPS depolymerases and virion associated (structural) lysins intervene at the beginning of the infection process to enable the phage to insert its DNA into the host bacterium[20,43,44] (cell wall depolymerase category in Fig. 1a). Endolysins have a completely different function, serving to degrade the bacterial cell wall rapidly at the end of the infection process to liberate the newly produced phage virions (in lysis category in Fig. 1a).

From a machine learning perspective, it is imperative that the functional groups we want to predict are consistent with the underlying biology. The PHROG categories were not originally intended as labels for ML; they contain significant overlap that certainly adds noise and likely impacts the performance of models trained on them. For example, DNA-associated proteins are included in 7 of the 9 PHROG categories: (1) "DNA, RNA and nucleotide metabolism", (2) "integration and excision", (3) "transcription regulation", (4) "moron, auxiliary metabolic gene and host takeover", (5) "head and packaging", (6) "tail" and (7) "other". This means that there are proteins with very similar functions (in the DNA-associated category) that are present in the positive and in the negative sets for all of these classes when using the PHROG categories as training labels.

In addition, very few biological insights are gained by predicting that a protein belongs to the other category as is done in VPF-PLM because (1) the specific annotation that is present in the PHROG

database is lost and (2) this category is simply an agglomeration of proteins with differing molecular functions. To name only a few, it is composed of methyltransferases, proteases, recombinases, kinases, lipoproteins, etc. The same can be said about the "moron, AMG and host takeover category" being composed of oxygenases, exclusion proteins, toxins, glucosyltransferases, ABC transporters, ribonucleotide reductases, etc.

Here, a reorganization of the PHROG categories into groups that share a common molecular function was realized in order to increase the accuracy, confidence and sensitivity of models. In many cases, this required creating a hierarchy that can be used to differentiate proteins that share a common high-level function but that perform different tasks (lower-level functions). This is the case for structural proteins (capsid, tail, collar, etc.), for proteins associated with lysis at the end of the infection cycle (lysin, holin, spanin, lysis inhibition), and for DNA-associated proteins (nuclease, integrase, transcriptional regulation, etc.).

Even though it is difficult to evaluate the generalization capabilities of such a model, we evaluated how our tool would behave on non-homologous proteins by testing it on proteins with no detectable similarities to proteins in the training set (see holdout procedure in the Methods section under 'Evaluation of robustness'). Empathi still performed well showing that even without any detectable similarities at the sequence level, protein embeddings still encode a signal that can be used to make predictions. Of course, as more proteins are discovered and added to databases, our model will likely need to be retrained to consider this new diversity.

Furthermore, the dataset used to train Empathi was annotated using PHROG pHMMs. This is currently the most well-adapted method as it can provide the specific (low-level) annotations required to train our models, but they still remain secondary annotations not verified experimentally that could potentially introduce biases in our models. Having a large diversity of protein sequences in each group helps to mitigate these biases. Removing clusters containing proteins with differing annotations from the training set also reduces potential noise. Still, a feature of our tool is to provide the confidence of each prediction. With this information, a user can choose the desired trade-off between high specificity (confidence threshold of 95-99%) and high sensitivity (default threshold of 50%). In this latter case, more predictions would be obtained at the expense of a higher false positive rate, which is acceptable if the alternative is not having any annotation at all. See Supplementary Note 1 for further analyses on potential false positives stemming from out-of-distribution proteins.

The PLM underlying all Empathi classification models, being trained on data from UniRef 45 and BFD[46] (Big Fantastic Database),

may not have an optimal representation of the entire viral universe. Consequently, in future works, fine-tuning the PLM on a database of metagenomically-sourced viral proteins may help it to better depict the large diversity of phage proteins and in turn to improve the functional classification task.

In conclusion, we believe this tool, relying on newly defined functional categories, constitutes an important step forward and will provide a more comprehensive view of the functions possessed by phages helping to better characterize and understand them.

## Methods

### Collecting and annotating phage proteins

Using INPHARED[36], 18,477 phage genomes were collected from GenBank[47] (Fig. 1b) along with their predicted protein sequences on January 2nd 2024. A significant proportion (82%) of these genomes correspond to dsDNA phages (Baltimore Group 1), but ssDNA phages (Baltimore Group II) are also present (15%). The remaining 3% are either uncharacterized or RNA viruses. A total of 1.85 M phage proteins predicted by Prokka[48] (which employs Prodigal[49]) were deduplicated (identity of 100%) into 904k unique proteins. To annotate them functionally, these proteins were compared with the pHMMs of the PHROG[33] database using HH-suite[50], the best hit being considered for each protein (e-value less than 0.001). Using ProtTrans[30], a protein language model learned on billions of tokens (amino acids), embeddings in the form of fixed-size 1024-dimensional vectors were computed for every protein in the dataset. Finally, proteins were clustered using MMseqs2[37] using a 30% sequence identity threshold, 80% coverage and an e-value less than 0.001 to create training and testing sets for machine learning.

### Training and testing models

Support vector machines (SVM) with an RBF (Radial Basis Function) kernel were used as the base classifier in all models. Logistic regression models and random forests were also evaluated during preliminary tests but drops in performance were observed in comparison to SVMs.

Empathi is composed of a set of binary classifiers (one per category as defined in Fig. 1a). As a result, it was necessary to define a new training and testing set for each functional category (Supplementary Table 1). This ensures that each model is trained on positive and negative data that is adapted to each category and that contains no overlap and as little noise as possible. It also means that the classifier for each category was both trained and tested independently from all other categories. The unknown category is intrinsic to Empathi as it corresponds to the case where all binary models return the negative class.

The training and testing set for each category was created as follows. Empathi is trained on the annotated portion of phage proteins obtained from INPHARED. Only clusters containing proteins with no contradictory annotations (not associated to both the positive and negative set) were kept to train models. Proteins with unknown functions, even if they are found in a cluster with other proteins with known functions, were not considered when training models. To constitute the positive class, all proteins assigned to this category were considered. Both to ensure a diverse selection of proteins in the negative class and to make positive and negative classes as homogenous as possible in size, only one protein per MMseqs2 cluster was incorporated in the negative class for general categories. For sub-categories, only proteins from the parent category are considered, allowing the model to focus on distinguishing the finer differences between similar proteins. In these cases, less proteins are available to constitute the negative set and therefore all proteins from the negative class (all proteins per cluster) were considered when training these models. For most functional categories, the negative class still contained 3–4 times more proteins than the positive one (Supplementary Table 1). To correct for this imbalance, weights on model per-class performance (1/ frequency of each class) were added during training. In some cases, proteins with very general annotations were left out of both positive and negative classes. For example, when training a model to predict tail proteins, proteins only annotated as phage virion proteins, i.e., the parent category, were excluded as these could correspond to tail or non-tail proteins. In particular, even if they are part of a separate category, some transcriptional regulators were included in the positive set for the DNA-associated model if their annotation allowed for it, while others were included in the negative set (some regulators activate or inhibit other proteins) or excluded from both the positive and negative sets if their annotations were too general. Finally, to limit data leakage between training and testing sets (i.e., similar proteins being found in both datasets, leading to an over-estimation of model performance), all proteins from one MMseqs2 cluster were either included in the training or in the testing set. For each category, the training set is built by randomly sampling 80% of clusters, leaving the remaining 20% to constitute the testing set.

Since models for sub-categories were trained using only the proteins in the parent category, a protein must first be predicted as being part of the parent category before being assigned to the sub-category. For example, a protein must first be predicted as being a PVP before it is assigned (or not) the tail annotation. As more data is used to train the parent classes, this approach results in higher confidences and sensitivity on general functions while allowing to assign specific functions when possible. Furthermore, this approach reduces the required computational resources as models of sub-categories are only applied on proteins assigned to their parent category.

Precision, sensitivity and F1-score were computed to measure the performance of each binary model and were reported as a function of the confidence of predictions.

### Evaluation of robustness

Next, we wanted to evaluate the robustness of our approach, in particular, to see how well our model can generalize to proteins that are very different from those seen during training. To this end, proteins from a given homologous group (a PHROG) and all proteins from similar PHROGs were removed from the training set (similarities between PHROGs being indicated in the PHROG database). Then, a model was trained, excluding these proteins, and its performance was evaluated on proteins corresponding to the holdout PHROGs. As a new model needs to be retrained for every iteration, this procedure is computationally intensive and was only performed ten times for six chosen categories (lysis, cell wall depolymerase, DNA-associated, PVP, transcriptional regulator and transferase).

### Annotating new datasets with Empathi

Firstly, three metaviromes were obtained from Garmaeva et al.[40] (gut), Bi et al.[38] (sulfuric soil), and Ni et al.[39] (marine water). Complete genomes were identified using CheckV[51] and two genomes were randomly chosen from each study. Proteins were predicted using Prodigal, embedded using ProtTrans and Empathi was used to predict their functions. Predictions were compared to the annotations obtained by PHROG pHMMs (HH-suite; best hit with e-value $< 1e^{-3}$), to the predictions made by VPF-PLM using their calibrated thresholds (--efam_calibration_threshold) and to predictions made using PHOLD[34], a structure-based annotation tool.

Secondly, the EnVhogDB[4] database composed of phage proteins obtained from metagenomic experiments was downloaded (on June 20th 2024) and protein functions were predicted with Empathi. No additional manipulations were required as protein embeddings for representative sequences of each EnVhogDB cluster together with their corresponding PHROG annotations were available. EnVhogDB clusters are created as described in Pérez-Bucio et al.[4] by using the pHMMs of sequence-level clusters at 30% identity.

Lastly, the proteins in the EFAM database[35] were downloaded along with their corresponding functional annotations obtained from DRAM[41]. Annotations obtained using metaproteomic data of virion-associated proteins were also downloaded from EFAM. Protein embeddings for all proteins were computed using ProtTrans and Empathi was used to predict their functions.

## Reporting summary

Further information on research design is available in the Nature Portfolio Reporting Summary linked to this article.

## Data availability

The data generated in this study have been deposited in the Zenodo archive available at https://doi.org/10.5281/zenodo.14036011. The EnVhog and EFAM databases are available at http://envhog.u-ga.fr/envhog/ and https://doi.org/10.25739/9vze-4143, respectively.

## Code availability

Empathi source code and code used for analyses can be found at https://huggingface.co/AlexandreBoulay/EmPATHi. Empathi can also be downloaded from https://pypi.org/project/empathi/ and as an Apptainer image from https://cloud.sylabs.io/library/alexandreboulay/empathi/empathi.

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

## Acknowledgements
A.B. is supported by fellowships from FRQNT (#325947) and RHHDS NSERC CREATE program. E.R. is funded by a Research Scholars—Junior 1 in artificial intelligence and digital health by FRQS (#307935). This research was enabled in part by support provided by Compute Ontario (https://www.computeontario.ca/) and the Digital Research Alliance of Canada (alliancecan.ca).

## Author contributions
A.B. and C.G. conceived the project. A.B. designed and implemented the computational framework. A.L. provided invaluable insight for the definition of functional categories. All authors contributed to the analysis and interpretation of the results. F.E. aided in building figures. E.R. and C.G. supervised the project. A.B. wrote the manuscript. All authors substantially revised the manuscript and approved the final version.

## Competing interests
The authors declare no competing interests.
