## [Transparent Peer Review file · Nature Communications]

Empathi: Embedding-based Phage Protein Annotation Tool by Hierarchical Assignment

Corresponding Author: Mr Alexandre Boulay

Version 0:

Reviewer comments:

Reviewer #1

(Remarks to the Author)

Boulay et al present Empathi, a tool that annotates phage genomes combining protein language model methods with more old school machine learning techniques (i.e. support vector machines) on top for classification. Overall, the manuscript is well written and of high quality. I also had no issues in installing and running the tool from the HuggingFace installation on my machine equipped with an RTX4090 GPU. It is open source and easy to install with all data on zenodo, for which the authors should be commended. Overall, they clearly have put a lot of thought and effort into designing and implementing empathi and it will be a valuable tool especially for annotating super difficult dark matter phage proteins. The hardest bit of benchmarking such work is to determine false positives, something I think the authors can and need to address in some more detail. My detailed comments follow.

Major Comments

Figure 1 i.e. Hierarchical Categories

The authors (some of whom are the same as those behind the PHROG database) reclassified the PHROG annotations and their 10 categories to a more hierarchical classification system that they say are more appropriate for machine learning purposes. Overall, I think what they have done in this regard is generally good with my one large caveat below, and certainly serves to advance their goals in a hierarchical classification tool, given the original 10 PHROG categories ranged from very broad (i.e. "Other") to quite specific ("connector"), making ML/DL methods harder to apply.

I have one main concern that is really my main issue with Empathi generally: quantifying false positives.

By not including a final catch-all "Other" or some such category to take into account proteins with rare or unusual (for phage functions), this likely reduces the ability of empathi to correctly label proteins that are outside of distribution of empathi' and crucially, increases false positives on such out of distribution proteins.

As an example, to test this, I took 500 random proteins from Swissprot and ran them through empathi. Empathi annotated 233/500 with a function. Most of these were quite high level less informative annotations ('DNA-associated'), though some of the more specific functions were indeed very reasonable if they were related to functions that phages and non-phages were likely to share – e.g. Q8DVB4 (Exodeoxyribonuclease 7 small subunit) is annotated as a 'DNA-associated|nuclease' and a number of transferase proteins seem correctly annotated. It is likely that the ProtTrans pLM used by Empathi understands proteins of this type quite well. However, the more phage-specific functions were commonly found as false positives – 8/500 were "cell wall depolymerases" including Q8DWI4 (Holliday junction branch migration complex subunit RuvB) and Q8DTC7 (Chaperone protein ClpB). 12/500 were annotated as 'pvp|tail', including O82713 (Small ribosomal subunit protein eS32) and Q45753 annotated as 'pvp|tail|tail_appendage' (Pesticidal crystal protein Cry5Ab) - proteins in these groups seemed all like clear false positives.

Figure 2 i.e. Dataset Curation & Test Sets

Why did the authors not use the PHROGs themselves for training, instead using INPHARED? I doubt it would make an

enormous difference to the model itself, but the PHROGs themselves have 868k proteins (compared to the 904k from INPHARED) so are the same size and are already annotated and could have made life easier downstream I think as it likely would have led to keeping more data given the fact that the authors excluded all proteins and clusters that were unannotated anyway (and removed all clusters with contradictory annotations).

Another comment here is that given their approach annotating phage genomes and using those for training, INPHARED is biased towards culturable phages that are commonly sequenced - the authors could have used other more diverse sources of phages (e.g. from PhageScope) too potentially.

Figure 3 i.e. Results Part 1 & Evaluation of Robustness

Figure 3 and accompanying results are nicely presented in combination with the Precision-Confidence and Recall-Confidence curves (and those in Supp 1). I am somewhat surprised to see very little variance in the curves between groups in Fig 3A even for the smaller groups (e.g. *crispr* or *sir2*) compared to the larger ones (e.g. PVP).

With Figure 3B, the authors say that the worse performance of these PHROGs are because the holdout PHROG is where all examples have been held out. Other than to say that this somewhat challenges the concept of transfer learning capabilities in pLMs generally, could the authors provide these a supplement? For something like 'transcriptional regulator' with many PHROGs and proteins in this category, one would think this concern should not necessarily hold.

Further, the authors state that 487k proteins (150k cluster) were not similar to any PHROG pHMM with a known function, and 300k were annotated with Empathi. This is impressive and this reviewer indeed believes Empathi can annotate (at least most of) these proteins, but questions whether all of these 487k proteins were really "dark". There are many 'unknown similarity' PHROGs with very strong HMM-HMM similarity to other annotated PHROGs (and were not given an annotation as such by the PHROGs team for various reasons) as presented in the PHROGs webserver, so I would suggest 300k/487k overstates the performance of Empathi (the denominator should be smaller). There were likely pHMM hits to "unknown function" PHROG groups (as the top hit) that can in reality be assigned a function from strong lower but not top hits. It might be better here to take the top 5 or 10 HMM-HMM hits and see whether there was a hit with a known function to reduce the 487k number to keep truly darker proteins.

Figure 4 Metagenomic Annotation

Overall this section looks quite good – the annotations look like they observed phage genome synteny quite well (albeit I imagine – the annotations are not included in the supps – that these are mostly quite high-level and relatively less useful annotations for the vast yellow and blue sections that constitute a lot of the difference i.e. I imagine these are a lot of 'DNA-associated' and 'pvp' that generally could be guessed from genome synteny alone anyway, but it is reassuring to see Empathi looks like it probably 'correct'). Based on my tests, I have no doubt Empathi should strongly outperform PHROGs HMM annotation and VPF-PLM as reported (which I have found to have similar performance as reported in Fig 4 in my hands i.e. not too much different to HMM annotation).

However, one type of comparison method is missing: structural. Given pLMs learn protein structural information (see e.g. the ESMfold paper and <https://doi.org/10.1073/pnas.2406285121>), it is likely that empathi is implicitly conducting some kind of structural motif -> function mapping in its models. Therefore, it would good to compare the performance of empathi vs explicit structural comparisons (e.g. with Foldseek vs AFDB or BFVD) – an example was conducted in the recent BFVD paper (<https://doi.org/10.1093/nar/gkae1119>) showing annotation rates in the same vicinity as empathi (Figure 3). While computationally expensive to generate e.g. Colabfold structures, I am sure this team has the capability and resources, and it would be good to conduct on the 6 phages in Figure 4 as a baseline. In particular, this reviewer would be interested in the proteins that Empathi can annotate but structure-search can't (and vice versa), as I imagine an interesting use case of empathi going forward is to provide some (even high level) guesses for function of the true "dark matter" ultra-tough to annotate proteins that even structure search may fail.

Section 3.3

I find the start of this section quite speculative and would advise caution in reassessing the claims. I truly doubt empathi can annotate 67.5% of the enVhogs accurately at the lower threshold of 50% without many many false positives and is a bit too speculative for my liking. This is one downside of empathi's metrics compared homology based search tools with well calibrated e-values – even 95% confidence implies 5% error (compared to e.g. an E-value of 0.001 used in HMM-HMM searches in this paper which implies a far lower rate) – the higher likely FP rate is one reason why I think empathi will be very useful for the community more as a tool to annotate the truly dark matter (where a high FP is more tolerable if the alternative is no information at all!) The more stringent claim later in the paragraph (33% annotation at 95% confidence) is a lot more reasonable.

Other Comments – explicitly, this reviewer would not require doing these things prior to publication acceptance, just more out of interest/to ask whether the authors have considered this

1. Why did the authors choose ProtT5 rather than e.g. ESM2 as their base pLM? Did they try similar experiments on ESM2, especially smaller model sizes (which would make embedding generation, the main resource requirement of Empathi, faster?)

2. Did the authors consider finetuning the underlying protein language model to better understand phage proteins? I imagine that the INPHARED proteins are indeed well understood by ProtTrans (1 because INPHARED phages will be captured reasonably well in UniRef given many are in RefSeq and 2 the metagenomic datasets trained on by ProtTrans should have some environmental phage proteins), but I do think that there may be gaps that would probably mean the pLM doesn't understand some proteins well – especially e.g. on very diverse enVhogs which buttresses my criticism above about the 67.5% claim above.

(Remarks on code availability)

Code installed with a pip installation as per the user instructions from Huggingface and ran correctly on my Ubuntu machine with RTX4090 GPU.

Reviewer #2

(Remarks to the Author)

Overview

The manuscript by Boulay et al. addresses the significant challenge of annotating viral protein sequences, specifically those from bacteriophages. This remains a persistent problem in viral genomics, with current methodologies only successfully annotating approximately 20% of phage proteins. The authors present a two-pronged approach to improve this situation: (1) the development of hierarchically organized protein categories (based on phrogs) to enhance clustering and annotation capabilities, and (2) the implementation of 44 distinct binary classification models corresponding to each protein category.

Strengths

The performance metrics for the developed models generally indicate reasonable efficiency. A particularly noteworthy advantage of this approach is the implementation of hierarchical classification categories, which provides a more organized framework for protein annotation than existing flat classification systems.

Major Concerns

Reference Dataset Limitations

The exclusive use of the INPHARED database represents a significant limitation of the current study. INPHARED exhibits known biases and does not adequately represent the diversity of phage genomes and, by extension, phage proteins. The INPHARED authors themselves acknowledge that "As of January 2021, 14,244 complete phage genomes have been sequenced. The INPHARED data set is dominated by phages that infect a small number of bacterial genera, with 75% of phages isolated on only 30 bacterial genera." A more comprehensive and representative dataset could have been derived from resources such as IMGVR or constructed de novo by the authors.

Training Dataset Characterization

Additional metadata regarding the training dataset composition is necessary for proper evaluation. Specifically, information about the inclusion of both dsDNA and ssDNA phages and the environmental distribution of source organisms would help assess the breadth of applicability of the resulting models.

Functional Category Assignment Logic

The rationale for assigning proteins to multiple functional categories requires clarification. The overlap between functional categories may introduce confusion in the classification system and potentially reduce its discriminatory power.

Model Performance Inconsistency

Some models demonstrate suboptimal F1 scores, particularly for protein categories such as collar proteins. A more selective approach that retains only high-performing models would enhance the overall reliability of the classification system.

Potential Overtraining and Data Leakage

The derivation of both training and testing datasets from INPHARED raises concerns about potential overtraining and data leakage. This concern is substantiated by the observed performance discrepancies when applying Empathi to metagenomic sequences, suggesting limited generalizability beyond the training data distribution.

Analytical Reporting Structure

The presentation of analyses combining "training set + testing set + unknown" datasets complicates the independent assessment of model performance across these distinct data categories. Separate evaluation and reporting for each dataset would provide clearer insights into model robustness and potential biases.

Hierarchical Information Organization

While the hierarchical approach represents an innovative aspect of this study and potentially enhances the biological relevance of annotations through regularization of specificity, the lack of an objective function for hierarchy construction limits the reproducibility and optimization of this organizational structure. Could you use contrastive loss, triplet loss, or ladder loss in training to improve this approach?

Limited Improvement in Annotation Capacity

The degree of improvement over existing annotation methodologies appears modest. The similarity between Empathi and PHROG ontologies, coupled with the relatively coarse granularity of approximately 40 annotation categories, suggests incremental rather than transformative advancement in phage protein annotation capabilities.

Minor concerns

Line 99: What does structure mean here?

Conclusion

While Empathi represents a step forward in addressing the challenges of phage protein annotation through its hierarchical classification approach, significant limitations in dataset selection, model training issues, and modest annotation improvements constrain its current utility. This study will benefit from incorporation of more diverse training data (including metagenomic sequences), refinement of models to exclude underperforming categories, development of more granular annotation capabilities, and more robust training with the use of objective functions.

(Remarks on code availability)

Software code

I found the installation rather difficult and messy with the pip install as it seems like the authors have not implemented argparse into the python script. I would really appreciate a conda install process. While the code seemed to work fine, it needed a bit of work to get the dependencies setup.

I also tested Empathi on a single phage and obtained output. For me the output formats were reasonable and I was able to understand the different columns. However, I highly recommend adding more details on the github page as these details about what each column means may not be obvious to most users.

Reviewer #3

(Remarks to the Author)

The study by Boulay et al. describes a new tool, Empathi, that helps with annotating bacteriophage (phage) genomes. The tool leverages a previously published protein language model (PLM) called ProtTrans to assign functions to phage proteins. The authors defined a hierarchy of phage functional categories and used these categories to train Empathi's classifiers. These classifiers are then used to assign, in a hierarchical manner, functions to some previously unannotated proteins from cultured and uncultured phages.

Strengths of the manuscript:

- The tool and the functional hierarchical scheme introduced by the authors are relevant and useful to the phage research community, especially in the era of metagenomics and viromics. On a selection of six genomes, the authors show that their tool outperforms homology-based annotation of phage genomes using the PHROG HMM database as well as the PLM-based annotation of these genomes using the methods of Flamholz et al. (hereafter called VPF-PLM)

Limitations of the manuscript:

- The manuscript would greatly benefit from major restructuring and some "Results" sections would benefit from major re-writing. I found it hard to understand the main messages and the steps that the authors followed except after re-reading the manuscript multiple times. For example, Supplementary figure 2 needs to be moved up to be one of the panels in Figure 1, so that the reader can follow what has been done early on. Another example, the authors start the "Building and testing models" section with "From the INPHARED dataset of phage genomes, 904k dereplicated proteins". The authors have not introduced "what" and "why" INPHARED dataset yet. This is a common theme across all the sections under "Results." The authors cannot assume that all their readers are bioinformaticians and/or knowledgeable about the different tools and datasets. Again, when the authors say, "Of the 198k clusters generated using MMseqs2," they leave the reader wondering "what clusters?"

- The documentation of the code ([huggingface.co/AlexandreBoulay/EmPATHi](https://github.com/AlexandreBoulay/EmPATHi)) would similarly benefit from more elaboration beyond the installation instructions and usage commands. Please provide samples of input, model, and output files. Additionally, please provide a readme file describing the files deposited under (zenodo.org/records/14036012)

- All Supplementary Tables lack captions and numbering. Please make sure to revise these against the citations within the text and provide sufficient descriptions of these Supplementary Tables.

- It was not clear to me whether the improvement in performance of Empathi over VPF-PLM was only due to VPF-PLM's PHROG categories possessing "overlapping molecular-level functions creating noise in the training data that hinders the accuracy and sensitivity that can be achieved by models trained on them." Empathi uses ProtTrans as a PLM, whereas VPF-PLM uses ESM2. Can the difference be attributed to just the underlying PLMs of the different tools? This is one of a few instances where I found the study not showing an apples-to-apples comparison.

- Another instance of where an apples-to-apples comparison should have been made is Figure 3. ProtTrans in Empathi sources from a much larger pool of protein sequences than PHROGs. What if the authors, instead of just relying on PHROG annotations, used a more comprehensive annotation tool such as DRAM that searches multiple databases? In fact, the EFAM paper showed that DRAM alone can double "the number of annotations obtainable by standard, single-database annotation approaches." This is similar to the authors' finding that "Empathi doubled the annotated fraction of protein families from 16% to 33%." What can be concluded from this regarding the improvements introduced by PLMs over large-scale, multi-database, homology-based annotation methods?

- On the mention of EFAM, perhaps there is a golden opportunity missed here to show the power of PLMs over homology-based methods. The EFAM database holds a large number of de novo annotated protein families using virion-associated metaproteomic data. Homolog-based methods failed to annotate these proteins. Can Empathi identify some of these de novo annotated proteins under any functional category in Empathi's hierarchy?

- Statements such as "phages have been overlooked in studies of the .." are now incorrect. There is a huge body of literature from JGI on phages in nature and animal systems. Please make sure to properly search and cite the literature in your introduction. Another incorrect statement is ".. only 16% of the diversity of phage protein families has been assigned a function." The EFAM database annotated well above 30% of its protein families.

- Please review for typos and grammar throughout the manuscript. Examples: "Using INPHARED, 18,477 phage genomes [with] were collected" and "is a multilabel classifier that uses embeddings obtained using ESM2 and [that is] trained to predict ..."

(Remarks on code availability)

Feedback is provided above under "Comments for Author"

Reviewer #4

(Remarks to the Author)

Empathi: Embedding-based Phage Protein Annotation Tool by Hierarchical Assignment, by Boulay et al., presents a novel, ML-based advance toward solving a persistent and vexing program in bacteriophage biology: reliable functional annotation of predicted encoded proteins. To do this, the authors have refactored the PHROG database categories in order to make them better suited for ML classification and created Empathi, the software tool and set of binary models that classifies using these new hierarchical categories. The benefit of this approach can be observed in comparison to Viral Protein Function prediction using Protein Language Model (VPF-PLM), which classifies phage proteins using PHROG categories. The authors provide some evidence to compare approaches, although I find it somewhat difficult to evaluate that evidence in a definitive way. The text on lines 213-216, for instance, highlights that Empathi annotated a higher proportion of proteins than VPF-PLM, but how are we to know that the difference isn't that VPF-PLM requires higher confidence to make a prediction? The Flamholz reference is of some use in evaluating this, but addressing the issue directly in this text would help a great deal to establish whether Empathi does indeed offer a substantial improvement. (I do believe that the authors have established that the refined PHROG categories are an improvement.)

In reading this article, I wonder about how accessible phage biologists may find the methods. Potential points of clarification that the authors could make to help their phage biologist readers who are not necessarily well-versed in ML terms and concepts: vectors, embeddings, data leakage, F1-scores.

On lines 323-324: some explanation of why these approaches were chosen is warranted. Were other approaches tried?

The methods and online documentation for Empathi are sufficiently detailed to allow for reproduction of the work.

Minor issues:

lines 25-26 refer to next-gen sequencing methods and shotgun sequencing as recent developments. Neither strikes me as recent, but particularly not shotgun sequencing.

Line 43: high mutation rates are a property of some viruses, but not all, and the range of mutation frequencies between virus families is enormous.

Line 47: functional homology. I prefer the classical definition of homology: descendant from a common ancestor. For this reason, I don't love the phrase "functional homology."

Line 251: "few" instead of "little"

Line 264: "with" instead of "to"

(Remarks on code availability)

I did not run the code, but it appears to be well organized and well documented both in terms of comments in the code and instructions on the website.

Version 1:

Reviewer comments:

Reviewer #1

(Remarks to the Author)

The authors have thoroughly addressed my comments overall, especially with regard to explaining false positive rates in this paper, where they have qualified and caveated some of their originally more aggressive statements with nuance and care. Two minor suggestions/comments:

1. (perhaps somewhat biased as I suggested it) but I really enjoyed the SwissProt analysis of out of distribution proteins presented in the rebuttal with manual/detailed annotation of "false positive" examples (with very nuanced interpretations of how much these can really be considered "false positive" or not i.e. they might not be phage proteins but have similarities to folds/domains found in phage proteins). This should be included as a supplementary note to this manuscript for interested readers.

2. Adding Phold (noting that it has a preprint now <https://www.biorxiv.org/content/10.1101/2025.08.05.668817v1>) to the analysis presented in Figure 3 is a good addition to that section, along with this sentence couching the predictions with some degree of caution. "However, a lower confidence is observed for 126/150 of these proteins, demonstrating that Empathi internally recognizes that these sequences are more distinct than those present in its training set."

(Remarks on code availability)

Handled in original review, code is open source and I tested installation

Reviewer #2

(Remarks to the Author)

I commend the authors for their thorough revisions and for responding to my queries. Based on these revisions, I think Empathi could become a widely used tool in the field. My only caveat is the lack of training on metagenomically sourced viruses that would have provided for a more balanced view of the viral diversity. I do recognize the limitations mentioned by the authors, and hopefully this can be improved upon in the future.

My only outstanding comment is to use higher default confidence thresholds. Providing the ability to modify confidence is great for advanced users but I fear that with higher confidence as a default, FPs may propagate in annotations.

(Remarks on code availability)

Code is reproducible. Some issues with installation that I had pointed out were fixed.

Reviewer #3

(Remarks to the Author)

No further comments. Thank you for the revisions and additional analyses.

(Remarks on code availability)

REVIEWER COMMENTS

Reviewer #1 (Remarks to the Author):

Boulay et al present Empathi, a tool that annotates phage genomes combining protein language model methods with more old school machine learning techniques (i.e. support vector machines) on top for classification. Overall, the manuscript is well written and of high quality. I also had no issues in installing and running the tool from the HuggingFace installation on my machine equipped with an RTX4090 GPU. It is open source and easy to install with all data on zenodo, for which the authors should be commended. Overall, they clearly have put a lot of thought and effort into designing and implementing empathi and it will be a valuable tool especially for annotating super difficult dark matter phage proteins. The hardest bit of benchmarking such work is to determine false positives, something I think the authors can and need to address in some more detail. My detailed comments follow.

Major Comments

Figure 1 i.e. Hierarchical Categories

Comment: The authors (some of whom are the same as those behind the PHROG database) reclassified the PHROG annotations and their 10 categories to a more hierarchical classification system that they say are more appropriate for machine learning purposes. Overall, I think what they have done in this regard is generally good with my one large caveat below, and certainly serves to advance their goals in a hierarchical classification tool, given the original 10 PHROG categories ranged from very broad (i.e. “Other”) to quite specific (“connector”), making ML/DL methods harder to apply.

I have one main concern that is really my main issue with Empathi generally: quantifying false positives.

By not including a final catch-all “Other” or some such category to take into account proteins with rare or unusual (for phage functions), this likely reduces the ability of empathi to correctly label proteins that are outside of distribution of empathi’ and crucially, increases false positives on such out of distribution proteins.

Response: We agree with the reviewer that an ‘Other’ category is important and necessary. Empathi is composed of binary classifiers (one for each function; i.e. the first model classifies Function 1 vs other, the second one function 2 vs other, etc.). The “other” category is hence intrinsic to the model as it corresponds to the negative class of each model. In the absence of a positive prediction by any of Empathi’s models, a protein is attributed the “unknown” label.

A clarification was added to the text at lines 391-392 and 112-113, and in figure 1.

Comment: As an example, to test this, I took 500 random proteins from Swissprot and ran them through empathi. Empathi annotated 233/500 with a function. Most of these were quite high

level less informative annotations ('DNA-associated'), though some of the more specific functions were indeed very reasonable if they were related to functions that phages and non-phages were likely to share – e.g. Q8DVB4 (Exodeoxyribonuclease 7 small subunit) is annotated as a 'DNA-associated|nuclease' and a number of transferase proteins seem correctly annotated. It is likely that the ProtTrans pLM used by Empathi understands proteins of this type quite well. However, the more phage-specific functions were commonly found as false positives – 8/500 were “cell wall depolymerases” including Q8DWI4 (Holliday junction branch migration complex subunit RuvB) and Q8DTC7 (Chaperone protein ClpB). 12/500 were annotated as 'pvp|tail', including O82713 (Small ribosomal subunit protein eS32) and Q45753 annotated as 'pvp|tail|tail_appendage' (Pesticidal crystal protein Cry5Ab) - proteins in these groups seemed all like clear false positives.

Response: We definitely agree that the hardest part is quantifying false positives. Figure 2a illustrates the false positive rate (FPR) of predictions on our testing set for the general categories. Already at a confidence level of 50%, models typically have a precision greater than 95%. See also supplementary table 1 for the precision scores of all models at 50%. Of course, as we increase the confidence threshold, the FPR should also improve. For these reasons, an option was added to Empathi to allow users to specify their desired confidence threshold.

Using proteins from Swissprot to test proteins that would be “out of distribution” compared to the phage proteins used to train Empathi was a great idea, and we thank the reviewer for this thorough investigation. We ran the same experiment in order to look at the confidence of predictions for such proteins. We also considered a subset of 500 randomly selected proteins including the 5 mentioned above, and we obtained very similar results. 287/500 were unknown and many (129) only had high-level functional annotations (e.g. DNA/RNA associated, transferase, transcriptional regulator, etc.). The most abundant specific annotations were integration-related (15) and nuclease (10).

As for phage specific annotations, 26/500 were annotated as PVPs. Most have low-moderate confidence (<0.80), but we will focus on the ones predicted with high confidence for the sake of this analysis.

- Q45753 that the reviewer identified has a confidence of 96%. Albeit not being a PVP, it is a protein containing the Galactose-binding domain (see Family and domains tab). This domain (IPR008979) is often found in phage tail proteins and receptor binding proteins (see the BFVD tab on the interpro page) for phages to adsorb to the bacterial cell wall surface, and likely to carbohydrates containing, among others, galactose <https://journals.asm.org/doi/full/10.1128/jb.188.7.2400-2410.2006>.
- C0HJH8 is a major capsid protein predicted with 99.7% confidence.
- A1V7N8 (confidence 90%) is a flagellar hook-basal body complex protein, so once again not a PVP, but a protein composing the structure of the flagellum of bacteria and acting as a connector.
- P0DOS2 (confidence 84%) plays a part in the formation of the virosome of the variola virus.

- P0C2W1 (confidence 82%) is a multifunctional protein found in many biological processes. The IPR013320 domain is found in tail fiber proteins and minor tail proteins (see the BFVD tab on the interpro page).
- Q84GK0 (confidence 82%) possesses many tailspike related domains including IPR012332 (pectate lyase) and IPR009003 (peptidase).

As a side note, O82713 that the reviewer mentioned had a confidence of 53%.

To summarise, although most of these proteins may not be PVPs, they share domains with PVPs and if they were found in phages, might adopt these roles.

13/500 were predicted as being cell wall depolymerases (degradation of sugars and peptides). Again, focusing only on high-confidence (i.e. above 80%) predictions, here is what we found:

- Q8DTC7 that the reviewer mentioned had a confidence of 95%. It is a chaperone and protease (PTHR11638). It is not a depolymerase, but chaperones are often part of tail fibers which assemble in trimeric structures to help with assembly.
- Q2YSD6 (confidence 86%) possesses the same protease domain as the previous protein.
- A0A1G9FQX8 (confidence 87%) is an adenosyl transferase and hydrolase. It was also predicted as being a transferase with 87% confidence.
- Q8AY81 (confidence 84%) has endopeptidase activity (IPR009003).
- Q2FWJ6 (confidence 96%). Contains the sprt-like domain (IPR006640), a protease found in autolysins of *S. pneumoniae*.
<https://pmc.ncbi.nlm.nih.gov/articles/PMC7939560/>.

As another side note, Q8DWI4 that the reviewer mentioned had a confidence of 59%.

All in all, this test, performed using proteins mostly unrelated to phages, shows that Empathi has a low false positive rate on proteins that are distinct from those seen during training. Additionally, most false positives identified possess domains usually found in phage tail proteins or depolymerases.

Comment: Figure 2 i.e. Dataset Curation & Test Sets

Why did the authors not use the PHROGs themselves for training, instead using INPHARED? I doubt it would make an enormous difference to the model itself, but the PHROGs themselves have 868k proteins (compared to the 904k from INPHARED) so are the same size and are already annotated and could have made life easier downstream I think as it likely would have led to keeping more data given the fact that the authors excluded all proteins and clusters that were unannotated anyway (and removed all clusters with contradictory annotations).

Response: We agree that the underlying source data (phage genomes from cultures) remains essentially consistent in both approaches. We developed a flexible scheme specifically to accommodate newly available phage data without being dependent on PHROGs release cycles. Using PHROGs directly as a fixed reference would require waiting for new official releases

before updating the Empathi training set. In contrast, using INPHARED allows us to readily update the training data by integrating proteins from any newly available phage genomes and annotating them with the current version of PHROGs (or eventually of integrating proteins from other sources). The performance of this scheme is therefore evaluated in this paper.

Comment: Another comment here is that given their approach annotating phage genomes and using those for training, INPHARED is biased towards culturable phages that are commonly sequenced - the authors could have used other more diverse sources of phages (e.g. from PhageScope) too potentially.

Response: We agree that training models on metagenomic data is the next logical step. However, doing so would require substantially more computational resources and would most likely necessitate certain adjustments to Empathi. The RAM required to train SVMs (complexity $O(N^2)$) is a strong limitation as well as the inference time which is proportional to the number of support vectors. In consequence, neural networks would probably be better suited for metagenomic data. In this case, retraining the protein language model (PLM) on environmental data (like EnVhog) may also be a good idea to better capture the added diversity of such metagenomic datasets. These are, in our opinion, the main ingredients for a future version of Empathi.

Comment: Figure 3 i.e. Results Part 1 & Evaluation of Robustness

Figure 3 and accompanying results are nicely presented in combination with the Precision-Confidence and Recall-Confidence curves (and those in Supp 1). I am somewhat surprised to see very little variance in the curves between groups in Fig 3A even for the smaller groups (e.g. crispr or sir2) compared to the larger ones (e.g. PVP).

Response: We believe that the diversity of the groups is more important than their size. As demonstrated in the holdout experiment (Figure 2b), lower performance tends to correspond to groups that contain subgroups of distinct homological origin.

A lower false positive rate (FPR) at a given confidence threshold likely reflects imbalances among these subgroups (i.e. the classifier may become biased toward the largest subgroup). Although the dereplication step helps to mitigate this issue, one potential improvement for future versions of Empathi could involve implementing sequence weighting that is inversely proportional to the size of each homological subgroup, which may help address this source of bias more effectively.

Comment: With Figure 3B, the authors say that the worse performance of these PHROGs are because the holdout PHROG is where all examples have been held out. Other than to say that this somewhat challenges the concept of transfer learning capabilities in pLMs generally, could the authors provide these a supplement? For something like 'transcriptional regulator' with many

PHROGs and proteins in this category, one would think this concern should not necessarily hold.

Response: A classifier cannot generalize to completely new proteins (with very different structures for instance), even though they may have the same function. Transcriptional regulators are a very broad category of proteins; some bind DNA while others activate or inhibit other proteins upstream in the biological pathway. Their molecular functioning may differ significantly (and thus their protein structures may as well) even though they have similar biological functions. If we remove a whole family of distinct regulators, there may not be other proteins with similar sequences/structures (and in turn embeddings) from which our models can learn to recognize them.

This test serves to push the boundaries of our classifiers. It demonstrates that for the most part, even if we remove all similar proteins – even those remotely related (as far as HMM-HMM comparisons can detect) – from the training data, our models remain consistent. For example, on average for the 6 categories presented in this analysis, the false positive rate is only 4.3% using the default confidence threshold of 50% and 2.2% using a confidence threshold of 95%. In our opinion, this hints at the generalization capabilities of PLMs for evolutionarily unrelated proteins.

Extra explanations were added at lines 155-164. Figure 2b was updated to include f1-score, precision and sensitivity of holdout iterations, highlighting that it is usually a drop in sensitivity, not precision that is observed.

Comment: Further, the authors state that 487k proteins (150k cluster) were not similar to any PHROG pHMM with a known function, and 300k were annotated with Empathi. This is impressive and this reviewer indeed believes Empathi can annotate (at least most of) these proteins, but questions whether all of these 487k proteins were really “dark”. There are many ‘unknown similarity’ PHROGs with very strong HMM-HMM similarity to other annotated PHROGs (and were not given an annotation as such by the PHROGs team for various reasons) as presented in the PHROGs webserver, so I would suggest 300k/487k overstates the performance of Empathi (the denominator should be smaller). There were likely pHMM hits to “unknown function” PHROG groups (as the top hit) that can in reality be assigned a function from strong lower but not top hits. It might be better here to take the top 5 or 10 HMM-HMM hits and see whether there was a hit with a known function to reduce the 487k number to keep truly darker proteins.

Response: We agree with the reviewer, and the numbers presented in the manuscript have been revised to take this into account (from lines 166-178). Only 4,046 proteins that had a first hit to an unknown PHROG had a hit to a second known PHROG (e-value < 0.001). An additional 2,444 proteins had a hit to an unknown PHROG that itself had a hit to a known PHROG (e-value < 0.001 and coverage \geq 80% between similar PHROGs). However, there were also 14k unannotated proteins that were present in 2.4k clusters containing other annotated proteins. Thus, 467.5k proteins (52%, previously 54%) and 148k clusters (75%, previously 76%) are truly unannotated (dark matter) before applying Empathi.

Comment: Figure 4 Metagenomic Annotation

Overall this section looks quite good – the annotations look like they observed phage genome synteny quite well (albeit I imagine – the annotations are not included in the supps – that these are mostly quite high-level and relatively less useful annotations for the vast yellow and blue sections that constitute a lot of the difference i.e. I imagine these are a lot of ‘DNA-associated’ and ‘pvp’ that generally could be guessed from genome synteny alone anyway, but it is reassuring to see Empathi looks like it probably ‘correct’). Based on my tests, I have no doubt Empathi should strongly outperform PHROGs HMM annotation and VPF-PLM as reported (which I have found to have similar performance as reported in Fig 4 in my hands i.e. not too much different to HMM annotation).

Response: We thank the reviewer for this careful investigation, and the confirmation of our results. Indeed, most predictions are high-level with 182 being PVP/tail proteins, 98 being DNA-associated, 32 packaging and 30 PVPs. However, many proteins with more specific annotations were also predicted (ex. 66 tail-appendages, 49 nucleases and 35 integration-related proteins, etc.). The decision to present only the most general annotation was made to increase the legibility of the figure. Empathi’s predictions for these genomes have been added to the Zenodo repository.

As a side note, evaluating Empathi predictions for proteins that did not receive annotations from other tools is harder. For these, we notice a drop in the confidence of our models, meaning that Empathi internally recognizes that they are more distinct from the training data. Consequently, they can be filtered using a higher confidence threshold (see lines 206-210).

Comment: However, one type of comparison method is missing: structural. Given pLMs learn protein structural information (see e.g. the ESMfold paper and <https://doi.org/10.1073/pnas.2406285121>), it is likely that empathi is implicitly conducting some kind of structural motif -> function mapping in its models. Therefore, it would good to compare the performance of empathi vs explicit structural comparisons (e.g. with Foldseek vs AFDB or BFVD) – an example was conducted in the recent BFVD paper (<https://doi.org/10.1093/nar/gkae1119>) showing annotation rates in the same vicinity as empathi (Figure 3). While computationally expensive to generate e.g. Colabfold structures, I am sure this team has the capability and resources, and it would be good to conduct on the 6 phages in Figure 4 as a baseline. In particular, this reviewer would be interested in the proteins that Empathi can annotate but structure-search can’t (and vice versa), as I imagine an interesting use case of empathi going forward is to provide some (even high level) guesses for function of the true “dark matter” ultra-tough to annotate proteins that even structure search may fail.

Response: As suggested, a comparison to a structure-based approach for functional annotation was added to the analysis presented in Figure 3 (genomic maps). Two tests were conducted:

- Foldseek was used in combination with the BFVD. Overall, the 574 proteins in the 6 genomes presented had a combined 18k hits to the BFVD, meaning that the same protein had hits to multiple proteins. However, only 251 proteins had hits to annotated proteins.

- Phold (<https://github.com/gbouras13/phold>), a recently developed tool that uses Foldseek to align protein structures to a database of over 1M predicted phage protein structures, was employed. Although the tool has not been published, it has two main advantages for our analysis: 1) the annotations are organized according to the PHROGs categories making them comparable to the other tools presented in the genomic maps and 2) only one functional annotation is given per protein thus avoiding the need to filter through thousands of annotations. A total of 263 proteins received an annotation. Results are consistent with what was described in our manuscript with the other tools. This analysis was added to lines 194-198 as well as a short description in the methods at line 450-451. Figure 3 was updated to include these new results.

Comment: Section 3.3

I find the start of this section quite speculative and would advise caution in reassessing the claims. I truly doubt empathi can annotate 67.5% of the enVhogs accurately at the lower threshold of 50% without many many false positives and is a bit too speculative for my liking. This is one downside of empathi's metrics compared homology based search tools with well calibrated e-values – even 95% confidence implies 5% error (compared to e.g. an E-value of 0.001 used in HMM-HMM searches in this paper which implies a far lower rate) – the higher likely FP rate is one reason why I think empathi will be very useful for the community more as a tool to annotate the truly dark matter (where a high FP is more tolerable if the alternative is no information at all!) The more stringent claim later in the paragraph (33% annotation at 95% confidence) is a lot more reasonable.

Response: We indeed agree on the fact that metagenomic-sourced proteins may be more distant and therefore FPR has to be better controlled. We reorganized this section, putting greater emphasis on the results at 95% confidence and explaining that although it is possible to get a greater number of predictions using a threshold of 50%, this also likely comes with a higher FPR. See changes in lines 239-247.

The holdout experiment presented in Figure 2B helps in the evaluation of FPs. Just as metagenomic sequences are more diverse and subsequently more evolutionarily distant from what the models have seen during training, the proteins in each holdout set are as well. The experiment demonstrates that the FPR on evolutionarily distant proteins is low, and that it is mainly sensitivity that is affected. Also, note that this test was performed using a 50% confidence threshold, rather than the 95% used with metagenomic sequences, hence demonstrating the high relevance of protein embeddings obtained from ProtT5 and our classification models to enhance protein annotations with low FPR.

The test using proteins from Swissprot (in a previous comment) is also a good measure of the FPR on evolutionarily distant proteins. It was seen that filtering proteins at 95% confidence is more than sufficient to filter out most potential FPs.

Comment: Other Comments – explicitly, this reviewer would not require doing these things prior to publication acceptance, just more out of interest/to ask whether the authors have considered this

1. Why did the authors choose ProtT5 rather than e.g. ESM2 as their base pLM? Did they try similar experiments on ESM2, especially smaller model sizes (which would make embedding generation, the main resource requirement of Empathi, faster?)

Response: We ran preliminary tests at the very beginning and found that ProtT5 led to better performances of our classification models than ESM2. This has been demonstrated by others previously: for viral protein homology detection, the performances on various pLM trained on large-scale data is comparable (see <https://openreview.net/forum?id=IEZjjDX0iC>). The same observation holds for other tasks, see (<https://doi.org/10.1371/journal.pone.0289030> Table 4) for the prediction of bacterial hosts, and (<https://www.nature.com/articles/s41598-025-86519-5.pdf>) for protein crystallization.

Comment: 2. Did the authors consider finetuning the underlying protein language model to better understand phage proteins? I imagine that the INPHARED proteins are indeed well understood by ProtTrans (1 because INPHARED phages will be captured reasonably well in UniRef given many are in RefSeq and 2 the metagenomic datasets trained on by ProtTrans should have some environmental phage proteins), but I do think that there may be gaps that would probably mean the pLM doesn't understand some proteins well – especially e.g. on very diverse enVhogs which buttresses my criticism above about the 67.5% claim above.

Response: We thank the reviewer for the suggestion. We agree on the fact that out-of-the-box pLM may not have an optimal representation of the entire viral universe, and we indeed considered fine-tuning a pLM on a diverse source of viral proteins. However, fine-tuning pLMs for viral proteins is a more fundamental and substantial work, relying on larger data and different techniques than what is at stake in the current manuscript. Given that our current tool brings substantial improvements over the state-of-the-art, we decided to leave it for further work. A paragraph was added in the discussion (see lines 358-362).

Comment: Reviewer #1 (Remarks on code availability):

Code installed with a pip installation as per the user instructions from Huggingface and ran correctly on my Ubuntu machine with RTX4090 GPU.

Response: We are very grateful to the reviewer for testing the installation and confirming that it could be completed successfully.

Reviewer #2 (Remarks to the Author):

Overview

The manuscript by Boulay et al. addresses the significant challenge of annotating viral protein sequences, specifically those from bacteriophages. This remains a persistent problem in viral genomics, with current methodologies only successfully annotating approximately 20% of phage proteins. The authors present a two-pronged approach to improve this situation: (1) the development of hierarchically organized protein categories (based on phrogs) to enhance clustering and annotation capabilities, and (2) the implementation of 44 distinct binary classification models corresponding to each protein category.

Strengths

The performance metrics for the developed models generally indicate reasonable efficiency. A particularly noteworthy advantage of this approach is the implementation of hierarchical classification categories, which provides a more organized framework for protein annotation than existing flat classification systems.

Major Concerns

Comment: Reference Dataset Limitations

The exclusive use of the INPHARED database represents a significant limitation of the current study. INPHARED exhibits known biases and does not adequately represent the diversity of phage genomes and, by extension, phage proteins. The INPHARED authors themselves acknowledge that "As of January 2021, 14,244 complete phage genomes have been sequenced. The INPHARED data set is dominated by phages that infect a small number of bacterial genera, with 75% of phages isolated on only 30 bacterial genera." A more comprehensive and representative dataset could have been derived from resources such as IMGVR or constructed de novo by the authors.

Response: We thank the reviewer for this suggestion. As mentioned previously in discussions with reviewer 1, training models on much bigger datasets is indeed possible. However, given the huge diversity of environmental data, it is expected that some proteins will be badly represented by the pLM. By training on "in distribution" sequences, we have more accurate models, although of course we lose in sensitivity. Training on metagenomic data would likely require fine-tuning the underlying pLM and switching to scalable base classifiers (as SVM training and inference do not scale well with a significant increase in the amount of data). This improvement is a great prospect for the next version of Empathi. For this version, we believe that the improvements we made compared to the state-of-the-art, especially the refined curation of the labels from PHROGs to constitute a hierarchical framework that is more adapted to machine learning tasks and more useful for biologists, and the gain in performances, already constitute advances.

Training Dataset Characterization

Comment: Additional metadata regarding the training dataset composition is necessary for proper evaluation. Specifically, information about the inclusion of both dsDNA and ssDNA phages and the environmental distribution of source organisms would help assess the breadth of applicability of the resulting models.

Response: Here is the composition of the training dataset: 82% of the phages downloaded to create the dataset correspond to dsDNA phages (Baltimore Group I). 15% correspond to ssDNA phages (Baltimore Group II). The remaining 3% correspond to uncharacterized or RNA viruses. This information was included in the methods (lines 369-372). For a next version of Empathi, utilizing metagenomic data would result in an increased diversity of the composition of the training data.

Unfortunately, no information on the environmental distribution of source organisms is available.

Comment: Functional Category Assignment Logic

The rationale for assigning proteins to multiple functional categories requires clarification. The overlap between functional categories may introduce confusion in the classification system and potentially reduce its discriminatory power.

Response: Empathi is composed of many binary classifiers (one per category in Figure 1). As a result, it was necessary to define a new training and testing set for each functional category (Supplementary table 1). This ensures that each model is trained on positive and negative data that contains no overlap and as little noise as possible. It also means that all models are independent from each other (i.e. not influenced by the proteins used to train other models). This allows for the same protein (ex. a tail spike protein) to be assigned multiple functions from 2 or more models (ex. PVP, tail and cell wall depolymerase).

A supplementary explanation was added in the Results at lines 114-115 and in the Material and Methods 386-392.

Comment: Model Performance Inconsistency

Some models demonstrate suboptimal F1 scores, particularly for protein categories such as collar proteins. A more selective approach that retains only high-performing models would enhance the overall reliability of the classification system.

Response: We agree with the reviewer. The collar protein model is still included in our analyses but has now been removed from the deployed version of Empathi as mentioned in the Results at lines 125-126.

Potential Overtraining and Data Leakage

The derivation of both training and testing datasets from INPHARED raises concerns about potential overtraining and data leakage. This concern is substantiated by the observed performance discrepancies when applying Empathi to metagenomic sequences, suggesting limited generalizability beyond the training data distribution.

Response: The clustering procedure used to create the testing set should minimize overtraining and data leakage, even if the data is derived from INPHARED. The main concern in properly evaluating models for this task is in testing them with proteins seen during training. Clustering proteins at 30% sequence identity ensures that proteins placed in the testing set are distinct from those used to train models. A sequence identity of 30% is already conservative in our opinion, but the additional evaluation performed using the holdout procedure described in the “Evaluation of robustness” section (starting at line 146) and employing HMM profiles should even more sensitively detect homologs. Additionally, using external datasets to evaluate the performance of models would lead to even more data leakage concerns as similar proteins would inevitably be found in these datasets, requiring the same clustering process to evaluate models.

It is true, as mentioned by other reviewers as well, that it is hard to evaluate false positives (FPs) that could arise as a consequence of the increased diversity present in metagenomic datasets. It is expected that there will be a limited generalization to unrelated sequences thus reducing the sensitivity of our models and increasing FPs. However, the results obtained using Empathi are still much better than annotations obtained from pHMMs (or the other tools presented).

The genomic maps in figure 3 serve to demonstrate this. We show that there is consistency between the predictions made by Empathi and the other tools for the ~260 proteins assigned a function by other tools. Of course, it is harder to evaluate the FP rate among the 151 proteins assigned a function by Empathi but not by the other tools. However, 126 of these 151 proteins have a lower confidence. Thus, Empathi internally recognizes that these sequences are more distinct than those present in its training set. More importantly, looking at the confidence of predictions allows users to filter proteins out depending on their goal (i.e. does the user want more annotated proteins at the risk of having more FPs, or does the user want more confident and more precise predictions, at the expense of having more false negatives). This extra analysis has been added in the Results at lines 206-210.

Following this line of thought, a confidence threshold of 95% was applied when analysing the EnVhogDB database simply out of precaution, to minimize the number of FPs. An increased confidence threshold results in Empathi making predictions for proteins that are more similar to those in its training set. We adopted a more modest tone when analysing the results on EnVhogDB at 50% and 95% confidence, reformulating to put more importance on the compromise to make between sensitivity and precision (see lines 240-247).

Also on this topic, reviewer 1 proposed an analysis on proteins obtained from Swissprot, corresponding mostly to proteins not seen during model training. We reran his analysis, taking 500 random proteins from Swissprot and predicted their functions with Empathi. To summarize, a bit less than half of the proteins received a prediction (213/500). Most corresponded either to general functions or functions likely to be found in Swissprot (for ex. DNA-binding proteins or nucleases). Some proteins were predicted as being phage virion proteins (26/500 with confidence >50%; 2/500 with confidence >95%) or depolymerases (13/500 with confidence >50%; 2/500 with confidence >95%). Most of these indeed correspond to false positives as they are not phage proteins, but most had low-moderate confidence (<80%), allowing them to be filtered out easily. Those predicted with higher confidence (>80%) possessed domains often found in phage tail proteins or depolymerases. Thus, this boosts our confidence in predictions made on proteins that are more distinct from those in the training set.

Finally, an option was added to Empathi to allow users to easily adjust the confidence threshold according to their objectives.

Comment: Analytical Reporting Structure

The presentation of analyses combining "training set + testing set + unknown" datasets complicates the independent assessment of model performance across these distinct data categories. Separate evaluation and reporting for each dataset would provide clearer insights into model robustness and potential biases.

Response: We thank the reviewer once again for this suggestion. These results were presented this way because our binary models were trained and tested separately - meaning that the data in the testing set of one model (e.g. the PVP model) can be found in the training set of another model (e.g. the DNA-associated model). This being said, we can still separate the previously unannotated clusters (i.e. the 'unknown' portion) from previously annotated ones. We will present those results here, but for the manuscript, we decided to add this type of analysis on the EFAM dataset instead. This case study was added following suggestions by another reviewer; it corresponds to a more well-known dataset and importantly has an "external" set of virion proteins identified using metaproteomic data. It allows us to compare Empathi to DRAM, another homology based annotation tool that makes use of multiple databases, and also to test Empathi's sensitivity for virion's proteins. The previous analysis combining "training set + testing set + unknown" datasets at lines 137-145 (in a strike-through font) was removed from the manuscript.

On the INPHARED dataset:

For the **49k clusters previously annotated by PHROGs**, 1) there were 20k singleton clusters, 2) all proteins in 25k clusters received completely identical annotations, 3) all proteins in 700 clusters received identical annotations or an unknown label, 4) all proteins in 3k clusters received at least one common annotation with secondary annotations that could differ and 5) 400 clusters contained proteins with differing annotations. For the **148k clusters not annotated by PHROGs**, 76k clusters are singleton clusters, 43k clusters received completely identical

annotations for all proteins in each cluster, 11k clusters received identical annotations or an unknown label, all proteins in 13k clusters received at least one common annotation with secondary annotations that could differ and 6k clusters contained proteins with differing annotations (analysis done using 50% confidence threshold).

On the EFAM dataset (new analysis at lines 257-288):

When looking at the **66,056 clusters previously annotated by DRAM** that also received predictions from Empathi, 51,856 clusters (79%) indeed received identical predictions for every protein. 1,600 clusters (2%) received identical predictions with some proteins receiving no function. The proteins in 12,201 clusters (18%) were predicted as sharing at least one general function but were assigned either differing specific functions or a differing second general function. Finally, only 399 clusters (<1%) received differing predictions. When looking at the **72,729 clusters only annotated by Empathi**, 44,669 clusters (61%) indeed received identical predictions for every protein. 9,995 clusters (14%) received identical predictions with some proteins receiving no function. The proteins in 17,138 clusters (24%) were predicted as sharing at least one general function but were assigned either differing specific functions or a differing second general function. Finally, only 927 clusters (1%) received differing predictions. There is thus a tendency in the previously unannotated portion of the dataset for clusters to possess more unknown predictions from Empathi as well. Most importantly, very few clusters possess proteins with differing annotations.

Comment: Hierarchical Information Organization

While the hierarchical approach represents an innovative aspect of this study and potentially enhances the biological relevance of annotations through regularization of specificity, the lack of an objective function for hierarchy construction limits the reproducibility and optimization of this organizational structure. Could you use contrastive loss, triplet loss, or ladder loss in training to improve this approach?

Response: The definition of these categories has been manually tailored based on biological expertise. Although there may be some way of discovering structure in the embedding space and to run unsupervised classification on it to get automatic labelling, it is much beyond the scope of this paper. For example, the suggestions of the reviewer could have great potential when developing a next version of Empathi trained on metagenomic data, where most of the data is unannotated, and where we would probably want to fine-tune the protein language model that we used so that it would be more specialized towards the diversity of virus proteins in metagenomic data.

Comment: Limited Improvement in Annotation Capacity

The degree of improvement over existing annotation methodologies appears modest. The similarity between Empathi and PHROG ontologies, coupled with the relatively coarse

granularity of approximately 40 annotation categories, suggests incremental rather than transformative advancement in phage protein annotation capabilities.

Response: While we agree that the Empathi annotation scheme shares ontological roots with PHROG ontologies, we would like to highlight that the primary aim of Empathi's annotation scheme is not to redefine the ontology itself, but to enhance the sensitivity of protein annotation tools by training them on categories that respect the molecular functions of proteins (less noise results in higher confidences and thus higher sensitivity). In this respect, we think the improvement is substantial: using Empathi, we were able to double the number of annotated orthologous groups in EnVhog compared to standard techniques (using a stringent confidence threshold of 95%). This is in line with the observations and conclusions of reviewer 1.

Future versions may indeed explore finer-grained categorization, but would require significantly more experimentally annotated data.

Comment: Minor concerns

Line 99: What does structure mean here?

Response: The “organisation” of the clusters. It has been modified (line 109).

Comment: Conclusion

While Empathi represents a step forward in addressing the challenges of phage protein annotation through its hierarchical classification approach, significant limitations in dataset selection, model training issues, and modest annotation improvements constrain its current utility. This study will benefit from incorporation of more diverse training data (including metagenomic sequences), refinement of models to exclude underperforming categories, development of more granular annotation capabilities, and more robust training with the use of objective functions.

Response: Responses to these remarks have been addressed previously.

Comment: Reviewer #2 (Remarks on code availability):

Software code

I found the installation rather difficult and messy with the pip install as it seems like the authors have not implemented argparse into the python script. I would really appreciate a conda install process. While the code seemed to work fine, it needed a bit of work to get the dependencies setup.

Response: Indeed, the installation from PyPI did not install the module as a callable script from bash (only from python). This has been fixed, making its usage much easier. We thank the reviewer for bringing this up.

Comment: I also tested Empathi on a single phage and obtained output. For me the output formats were reasonable, and I was able to understand the different columns. However, I highly recommend adding more details on the github page as these details about what each column means may not be obvious to most users.

Response: An example input and output file has been added to the code repository so users can easily test their installation. The format of the output file has been described on the GitHub page. Once again, we thank the reviewer for this very helpful comment.

Reviewer #3 (Remarks to the Author):

The study by Boulay et al. describes a new tool, Empathi, that helps with annotating bacteriophage (phage) genomes. The tool leverages a previously published protein language model (PLM) called ProtTrans to assign functions to phage proteins. The authors defined a hierarchy of phage functional categories and used these categories to train Empathi's classifiers. These classifiers are then used to assign, in a hierarchical manner, functions to some previously unannotated proteins from cultured and uncultured phages.

Strengths of the manuscript:

- The tool and the functional hierarchical scheme introduced by the authors are relevant and useful to the phage research community, especially in the era of metagenomics and viromics. On a selection of six genomes, the authors show that their tool outperforms homology-based annotation of phage genomes using the PHROG HMM database as well as the PLM-based annotation of these genomes using the methods of Flamholz et al. (hereafter called VPF-PLM)

Limitations of the manuscript:

Comment: The manuscript would greatly benefit from major restructuring and some "Results" sections would benefit from major re-writing. I found it hard to understand the main messages and the steps that the authors followed except after re-reading the manuscript multiple times. For example, Supplementary figure 2 needs to be moved up to be one of the panels in Figure 1, so that the reader can follow what has been done early on. Another example, the authors start the "Building and testing models" section with "From the INPHARED dataset of phage genomes, 904k dereplicated proteins". The authors have not introduced "what" and "why" INPHARED dataset yet. This is a common theme across all the sections under "Results." The authors cannot assume that all their readers are bioinformaticians and/or knowledgeable about the different tools and datasets. Again, when the authors say, "Of the 198k clusters generated using MMseqs2," they leave the reader wondering "what clusters?"

Response: We thank the reviewer for these helpful comments to enhance the readability of our manuscript. Supplementary figure 2 has now been placed as panel B in figure 1. The paragraph referred to here in the Results section was reorganized to explain the dataset creation better

(lines 101-110) as well as the next paragraph that explains the training of binary models (lines 112-120).

Comment: The documentation of the code (huggingface.co/AlexandreBoulay/EmPATHi) would similarly benefit from more elaboration beyond the installation instructions and usage commands. Please provide samples of input, model, and output files. Additionally, please provide a readme file describing the files deposited under (zenodo.org/records/14036012)

Response: An example input and output file has been added to the code repository so users can easily test their installation. The format of the output file has been described on the GitHub page. A README was added to the Zenodo repository.

Comment: All Supplementary Tables lack captions and numbering. Please make sure to revise these against the citations within the text and provide sufficient descriptions of these Supplementary Tables.

Response: We will ensure to fix this during the next submission (the submission portal did not include the titles and descriptions submitted along with supplementary material).

Comment: It was not clear to me whether the improvement in performance of Empathi over VPF-PLM was only due to VPF-PLM's PHROG categories possessing "overlapping molecular-level functions creating noise in the training data that hinders the accuracy and sensitivity that can be achieved by models trained on them." Empathi uses ProtTrans as a PLM, whereas VPF-PLM uses ESM2. Can the difference be attributed to just the underlying PLMs of the different tools? This is one of a few instances where I found the study not showing an apples-to-apples comparison.

Response: For viral protein homology detection, the performances on various pLM trained on large-scale data is comparable (see <https://openreview.net/forum?id=IEZjjDX0iC>). The same observation holds for other tasks, see (<https://doi.org/10.1371/journal.pone.0289030> Table 4) for the prediction of bacterial hosts, and (<https://www.nature.com/articles/s41598-025-86519-5.pdf>) for protein crystallization.

However, in VPF-PLM, the training set includes overlap between groups (i.e. similar proteins found in many groups), which is likely to perturb the classifier. For example, when looking at the "DNA, RNA and nucleotide metabolism" category, many DNA-binding proteins are part of other categories and are thus included in the negative instances for this class. These include DNA methyltransferases, integrases/recombinases, transcriptional regulators and DNA packaging chaperones which were included in categories such as "other", "integration and excision", "transcription regulation" and "head and packaging" instead. The presence of such proteins as negative instances during model training clearly hinders the classifier accuracy. Another related problem consists of proteins with multiple functions being assigned a unique label. Indeed, virion associated lysins are only classified as PVPs, despite sharing functional domains with endolysins (lysis-associated proteins). The resulting classifier thus loses critical information by predicting these proteins as PVPs and failing to detect the depolymerase function. The same is

true of many other proteins like the DNA methyltransferases mentioned previously that would just be classified as “other” using the PHROG categories as training labels.

The fact that we devised a scheme with the possibility of having several labels per protein helps to provide a clean training set to the classifiers. Proteins with a common molecular function are grouped together (i.e. all DNA-binding proteins are included as positive instances for the DNA-associated model), helping the model identify the common functional signal shared by these proteins. This, often achieved by defining more precise categories, also helps in reducing the heterogeneity observed within PHROG categories: there are many evolutionarily and structurally unrelated proteins inside PHROG categories, especially the “other” and “moron auxiliary metabolic gene and host takeover” categories. This trend diminishes as categories are constrained (defined more precisely) and consequently, embeddings tend to colocalize for these finer categories. Another advantage of having several labels per protein, besides improving model performance, is enabling the assignment of more than one function per protein. Using previous examples, virion associated lysins can be classified as both PVPs and depolymerases, and DNA methyltransferases can be classified as DNA-associated and transferases, which is much more useful from a biological perspective.

Finally, we tried several machine learning classifiers (Random Forest, Logistic Regression and SVMs) and SVMs worked best, leading to a marginal improvement here again. This classifier (SVM with an RBF kernel) makes a lot of sense since it naturally recognizes proximity to labeled shape-free patches in the latent space. The use of binary SVMs constitutes the other main difference with VPF-PLM.

Comment: Another instance of where an apples-to-apples comparison should have been made is Figure 3. ProtTrans in Empathi sources from a much larger pool of protein sequences than PHROGs. What if the authors, instead of just relying on PHROG annotations, used a more comprehensive annotation tool such as DRAM that searches multiple databases? In fact, the EFAM paper showed that DRAM alone can double “the number of annotations obtainable by standard, single-database annotation approaches.” This is similar to the authors’ finding that “Empathi doubled the annotated fraction of protein families from 16% to 33%.” What can be concluded from this regarding the improvements introduced by PLMs over large-scale, multi-database, homology-based annotation methods?

Response: We thank the reviewer for these great recommendations. Instead of launching DRAM on the 6 genomes in Figure 3, we decided to include a supplementary section on the EFAM dataset, allowing us to compare DRAM to Empathi on a dataset more people are familiar with. Additionally, it allowed us to perform the analysis the reviewer suggested in the next comment, which is equally interesting.

Of note, using a confidence threshold of 95%, Empathi annotated 57.7% (138,785 / 240,311) of clusters in the EFAM dataset compared to the 33.5% obtained from DRAM. A little less than 73k clusters were only annotated by Empathi, 14k were only annotated by DRAM, 66k were annotated by both and 87k clusters remained unannotated.

Next, we evaluated the consistency of Empathi predictions on previously annotated and unannotated clusters. When looking at the **66,056 clusters previously annotated by DRAM** that also received Empathi predictions, 51,856 clusters (79%) indeed received identical predictions for every protein. 1,600 clusters (2%) received identical predictions with some proteins receiving no function. The proteins in 12,201 clusters (18%) were predicted as sharing at least one general function but were assigned either differing specific functions or a differing second general function. Finally, only 399 clusters (<1%) received differing predictions. When looking at the **72,729 clusters only annotated by Empathi**, 44,669 clusters (61%) indeed received identical predictions for every protein. 9,995 clusters (14%) received identical predictions with some proteins receiving no function. The proteins in 17,138 clusters (24%) were predicted as sharing at least one general function but were assigned either differing specific functions or a differing second general function. Finally, only 927 clusters (1%) received differing predictions. There is thus a tendency in the previously unannotated portion of the dataset for clusters to possess more unknown predictions from Empathi as well. Most importantly, very few clusters possess proteins with differing annotations.

Finally, the proteins in EFAM annotated using virion-associated metaproteomic data were used to validate the predictions made by Empathi. Out of a total 29,355 of these proteins, 29,036 (99%) were predicted as being PVPs with a confidence >95% and 319 as being non-PVPs.

This is presented as a new section in the Results at lines 257-288.

Comment: On the mention of EFAM, perhaps there is a golden opportunity missed here to show the power of PLMs over homology-based methods. The EFAM database holds a large number of de novo annotated protein families using virion-associated metaproteomic data. Homolog-based methods failed to annotate these proteins. Can Empathi identify some of these de novo annotated proteins under any functional category in Empathi's hierarchy?

Response: Please refer to the previous comment.

Comment: Statements such as “phages have been overlooked in studies of the ..” are now incorrect. There is a huge body of literature from JGI on phages in nature and animal systems.

Response: We thank the reviewer for mentioning this as it allowed us to clarify. Here, we were not suggesting that there is no body of literature on phages, but rather that there is still a lot of work to do on phages to understand them better, especially in comparison to bacteria, which have historically been the main focus of most studies (see lines 22-23).

Comment: Please make sure to properly search and cite the literature in your introduction. Another incorrect statement is “.. only 16% of the diversity of phage protein families has been assigned a function.” The EFAM database annotated well above 30% of its protein families.

Response: This statement refers to proteins in the EnVhog database which encompasses a much greater diversity of proteins than those in EFAM. It includes proteins obtained from the genomes in EFAM, PHROG, RefSeqVirus, IMG/VR and GL-UVAB. The sentence was reformulated to make it clearer that we were referring to this database (see lines 32-33).

Comment: Please review for typos and grammar throughout the manuscript. Examples: “Using INPHARED, 18,477 phage genomes [with] were collected” and “is a multilabel classifier that uses embeddings obtained using ESM2 and [that is] trained to predict ...”

Response: Thank you for these two observations. They have been corrected, and the whole manuscript has been revised.

Reviewer #3 (Remarks on code availability):

Feedback is provided above under "Comments for Author"

Reviewer #4 (Remarks to the Author):

Comment: Empathi: Embedding-based Phage Protein Annotation Tool by Hierarchical Assignment, by Boulay et al., presents a novel, ML-based advance toward solving a persistent and vexing program in bacteriophage biology: reliable functional annotation of predicted encoded proteins. To do this, the authors have refactored the PHROG database categories in order to make them better suited for ML classification and created Empathi, the software tool and set of binary models that classifies using these new hierarchical categories. The benefit of this approach can be observed in comparison to Viral Protein Function prediction using Protein Language Model (VPF-PLM), which classifies phage proteins using PHROG categories. The authors provide some evidence to compare approaches, although I find it somewhat difficult to evaluate that evidence in a definitive way. The text on lines 213-216, for instance, highlights that Empathi annotated a higher proportion of proteins than VPF-PLM, but how are we to know that the difference isn't that VPF-PLM requires higher confidence to make a prediction? The Flamholz reference is of some use in evaluating this, but addressing the issue directly in this text would help a great deal to establish whether Empathi does indeed offer a substantial improvement. (I do believe that the authors have established that the refined PHROG categories are an improvement.)

Response: We thank the reviewer for pointing this out. VPF-PLM in fact uses confidence thresholds calibrated for false positives to make its predictions. Rather counterintuitively, the thresholds for the most abundant classes were decreased following this calibration. This implies, for example, that for a protein to be assigned to the “DNA, RNA and nucleotide metabolism” category, it must just receive a confidence greater than 25%. This is in stark contrast to the 95% confidence that was used with Empathi for this analysis. Here is a list of the calibrated thresholds used by VPF-PLM:

- DNA, RNA and nucleotide metabolism: 0.25
- transcriptional regulation: 0.29
- head and packaging: 0.41
- tail: 0.45
- integration and excision: 0.46
- other: 0.50
- lysis: 0.53

- moron, auxiliary metabolic gene and host takeover: 0.64
- connector: 0.74

Note that the two categories with the highest confidence thresholds are two of the smallest categories in terms of number of proteins, while the four categories with the lowest thresholds are the biggest categories.

These extra details were added to the text at lines 252-256.

Comment: In reading this article, I wonder about how accessible phage biologists may find the methods. Potential points of clarification that the authors could make to help their phage biologist readers who are not necessarily well-versed in ML terms and concepts: vectors, embeddings, data leakage, F1-scores.

Response: We thank the reviewer for pointing this out, it is indeed important to provide explanations for technical terms to make the manuscript more easily accessible to all potential readers and users of our tool. Precision for data leakage was given at lines 418-419. Extra explanations were added for precision, sensitivity and F1-score at lines 116-120. A definition of 'embeddings' was already provided in the introduction ('fixed-size real-valued vectors obtained from protein language models (PLMs)'), that we believe and hope is a brief, but clear definition and explanation, and we referred to figure 1B to give a visual example of what embeddings – and vectors – are (see lines 47-48). Also, the terms recall and sensitivity were used interchangeably. These were homogenized to use sensitivity everywhere which may be more familiar to biologists, with a description given at line 65.

Comment: On lines 323-324: some explanation of why these approaches were chosen is warranted. Were other approaches tried?

Response: Logistic regression models and random forests were also evaluated during preliminary tests but drops in performance were observed in comparison to SVMs (see lines 383-385).

Technically, any model can be used (logistic regression, random forests, neural networks, etc.), but support vector machines (SVMs) are particularly well adapted for embeddings. This has to do with the way SVMs work and the nature of protein embeddings. SVMs maximize the margin between different classes (distance from the decision boundary) and are capable of handling large feature spaces, effectively capturing non-linear relationships within the embedding space. Knowing that embeddings of similar proteins are closer together in the embedding space than to other proteins, an algorithm that maximizes the margin between groups should be well-suited both to obtain a good sensitivity and to minimize false positives.

Comment: The methods and online documentation for Empathi are sufficiently detailed to allow for reproduction of the work.

Response: We thank the reviewer for their validation of the methods and documentation.

Minor issues:

Comment: lines 25-26 refer to next-gen sequencing methods and shotgun sequencing as recent developments. Neither strikes me as recent, but particularly not shotgun sequencing.

Response: We removed “recent”.

Comment: Line 43: high mutation rates are a property of some viruses, but not all, and the range of mutation frequencies between virus families is enormous.

Response: We simplified the sentence so that there is no confusion, although the mutation rate does not need to be high in all viruses to contribute to the overall important diversity of proteins (see lines 43-46).

Comment: Line 47: functional homology. I prefer the classical definition of homology: descendant from a common ancestor. For this reason, I don’t love the phrase “functional homology.”

Response: This sentence was simplified to remove “and leverage functional homology between proteins”, which was not essential to the paragraph anyways (see lines 47-49).

Comment: Line 251: “few” instead of “little”

Line 264: “with” instead of “to”

Response: We thank the reviewer for noticing these grammatical mistakes, they have been fixed.

Comment: Reviewer #4 (Remarks on code availability):

I did not run the code, but it appears to be well organized and well documented both in terms of comments in the code and instructions on the website.

Response: We thank the reviewer for the revision of the documentation of our code repository.

Empathi: Embedding-based Phage Protein Annotation Tool by Hierarchical assignment.

Rebuttal letter 2

REVIEWERS' COMMENTS

Reviewer #1 (Remarks to the Author):

The authors have thoroughly addressed my comments overall, especially with regard to explaining false positive rates in this paper, where they have qualified and caveated some of their originally more aggressive statements with nuance and care. Two minor suggestions/comments:

1. (perhaps somewhat biased as I suggested it) but I really enjoyed the SwissProt analysis of out of distribution proteins presented in the rebuttal with manual/detailed annotation of “false positive” examples (with very nuanced interpretations of how much these can really be considered “false positive” or not i.e. they might not be phage proteins but have similarities to folds/domains found in phage proteins). This should be included as a supplementary note to this manuscript for interested readers.

Response: This analysis has been added as a supplementary note.

2. Adding Phold (noting that it has a preprint now <https://www.biorxiv.org/content/10.1101/2025.08.05.668817v1>) to the analysis presented in Figure 3 is a good addition to that section, along with this sentence couching the predictions with some degree of caution. “However, a lower confidence is observed for 126/150 of these proteins, demonstrating that Empathi internally recognizes that these sequences are more distinct than those present in its training set.”

Response: The reference for PHOLD was updated.

Reviewer #1 (Remarks on code availability):

Handled in original review, code is open source and I tested installation

Reviewer #2 (Remarks to the Author):

I commend the authors for their thorough revisions and for responding to my queries. Based on these revisions, I think Empathi could become a widely used tool in the field. My only caveat is the lack of training on metagenomically sourced viruses that would have provided

for a more balanced view of the viral diversity. I do recognize the limitations mentioned by the authors, and hopefully this can be improved upon in the future.

My only outstanding comment is to use higher default confidence thresholds. Providing the ability to modify confidence is great for advanced users but I fear that with higher confidence as a default, FPs may propagate in annotations.

Response: The default confidence threshold was modified to be 95%.

Reviewer #2 (Remarks on code availability):

Code is reproducible. Some issues with installation that I had pointed out were fixed.

Reviewer #3 (Remarks to the Author):

No further comments. Thank you for the revisions and additional analyses.

Response: We thank the reviewers once again for their recommendations.